# IL-27 maintains cytotoxic Ly6C$^+$ γδ T cells that arise from immature precursors

Robert Wiesheu [1,2], Sarah C Edwards[1,2], Ann Hedley [2], Holly Hall [2], Marie Tosolini [3], Marcelo Gregorio Filho Fares da Silva[4], Nital Sumaria[5], Suzanne M Castenmiller[6,7,8], Leyma Wardak[6,7,8], Yasmin Optaczy [2], Amy Lynn[1,2], David G Hill[9], Alan J Hayes[10], Jodie Hay[1], Anna Kilbey [1,2], Robin Shaw[2], Declan Whyte[2], Peter J Walsh[2], Alison M Michie [1], Gerard J Graham[10], Anand Manoharan[1], Christina Halsey [1], Karen Blyth [1,2], Monika C Wolkers [6,7,8], Crispin Miller[1,2], Daniel J Pennington [5], Gareth W Jones [9], Jean-Jacques Fournie[3], Vasileios Bekiaris [4] & Seth B Coffelt [1,2✉]

## Abstract

In mice, γδ-T lymphocytes that express the co-stimulatory molecule, CD27, are committed to the IFNγ-producing lineage during thymic development. In the periphery, these cells play a critical role in host defense and anti-tumor immunity. Unlike αβ-T cells that rely on MHC-presented peptides to drive their terminal differentiation, it is unclear whether MHC-unrestricted γδ-T cells undergo further functional maturation after exiting the thymus. Here, we provide evidence of phenotypic and functional diversity within peripheral IFNγ-producing γδ T cells. We found that CD27$^+$Ly6C$^-$ cells convert into CD27$^+$Ly6C$^+$ cells, and these CD27$^+$Ly6C$^+$ cells control cancer progression in mice, while the CD27$^+$Ly6C$^-$ cells cannot. The gene signatures of these two subsets were highly analogous to human immature and mature γδ-T cells, indicative of conservation across species. We show that IL-27 supports the cytotoxic phenotype and function of mouse CD27$^+$Ly6C$^+$ cells and human Vδ2$^+$ cells, while IL-27 is dispensable for mouse CD27$^+$Ly6C$^-$ cell and human Vδ1$^+$ cell functions. These data reveal increased complexity within IFNγ-producing γδ-T cells, comprising immature and terminally differentiated subsets, that offer new insights into unconventional T-cell biology.

**Keywords** Cancer; Differentiation; γδ T Cells; IL-27; Innate
**Subject Categories** Cancer; Immunology

## Introduction

Mouse IFNγ-producing γδ T cells expressing Vγ1 or Vγ4 T-cell receptors (TCRs) are migratory cells that travel between peripheral organs and secondary lymphoid organs (Ribot et al, 2021). These cells express the co-stimulatory molecule, CD27, which is absent from mature IL-17-producing γδ T cells (Ribot et al, 2009). Despite their low abundance, CD27$^+$ IFNγ-producing γδ T cells provide significant protection from pathogens and cancer (Ribot et al, 2021; Silva-Santos et al, 2019). CD27$^+$ IFNγ-producing γδ T cells defend against viral infection via directly killing infected cells (Khairallah et al, 2015; Mantri and St John, 2019; Sell et al, 2015). These cells counteract cancer progression in mouse models by killing cancer cells with granzymes and perforin as well as upregulating MHC-I expression on cancer cells to increase CD8$^+$ T-cell recognition (Dadi et al, 2016; Gao et al, 2003; Lanca et al, 2013; Riond et al, 2009). In addition to their endogenous anti-tumor role, CD27$^+$ IFNγ-producing γδ T cells control tumor growth after ex vivo expansion and adoptive cell transfer into tumor-bearing mice (Beck et al, 2010; Cao et al, 2016; He et al, 2010; Liu et al, 2008; Street et al, 2004), mirroring the outcomes of experiments using human Vγ9Vδ2$^+$ or Vδ1$^+$ cells (Silva-Santos et al, 2019). Within the CD27$^+$ IFNγ-producing γδ T-cell subset, there appears to be some nuance, as Vγ4$^+$ cells are better at restraining B16 melanoma tumors than Vγ1$^+$ cells (He et al, 2010). Additional subsets of IFNγ-producing γδ T cells have been identified independent of their TCR usage that can be distinguished by expression of the myeloid cell-associated molecule, Ly6C (Lombes et al, 2015); although the relationship between these subsets and their functional importance is unclear.

Several molecules and pathways regulate CD27$^+$ IFNγ-producing γδ T cells. Cytokines, including IL-2, IL-12, and IL-15, activate T-bet and EOMES transcription factors to drive expression

[1]School of Cancer Sciences, College of Medical Veterinary and Life Sciences, University of Glasgow, Glasgow, UK. [2]Cancer Research UK Scotland Institute, Glasgow, UK. [3]Cancer Research Centre of Toulouse, University of Toulouse, Toulouse, France. [4]Department of Health Technology, Technical University of Denmark, Kongens Lyngby, Denmark. [5]Blizard Institute, Barts and The London School of Medicine and Dentistry, Queen Mary University of London, London, UK. [6]Landsteiner Laboratory, Amsterdam UMC, University of Amsterdam, Amsterdam, The Netherlands. [7]Department Of Hematopoiesis, Sanquin Research, Amsterdam, The Netherlands. [8]Oncode Institute, Utrecht, The Netherlands. [9]School of Cellular and Molecular Medicine, University of Bristol, Bristol, UK. [10]School of Infection & Immunity, University of Glasgow, Glasgow, UK. ✉E-mail: Seth.Coffelt@glasgow.ac.uk

of IFNγ in these cells (Barros-Martins et al, 2016; Chen et al, 2007; He et al, 2010; Lino et al, 2017; Yang et al, 2020; Yin et al, 2002; Yin et al, 2000), while IL-2, IL-15, and IL-18 stimulate their proliferation (Corpuz et al, 2016; da Mota et al, 2020; Ribot et al, 2012). Co-stimulation through CD27 and CD28 augments survival and mitogenic signals for these cells (Ribot et al, 2010; Ribot et al, 2012). CD27$^+$ IFNγ-producing γδ T cells utilize glycolytic metabolism for energy, and glucose enhances their anti-tumor activity via mTOR regulation of T-bet, EOMES, and NKG2D expression (Lopes et al, 2021; Yang et al, 2020). By contrast, hypoxia suppresses the anti-tumor ability of these cells via HIF-1α down-regulation of IFNγ and NKG2D (Park et al, 2021).

We recently generated a γδ T-cell scRNAseq dataset from the lungs of naive mice, which indicated the possibility of increased diversity within CD27$^+$ IFNγ-producing γδ T cells. We identified two clusters of CD27$^+$ γδ T cells that expressed either genes associated with lymphocyte migration (*Ccr7* and *S1pr1*) or *Ly6c2* (Edwards et al, 2023). This observation is supported by other γδ T-cell scRNAseq datasets from other tissues (du Halgouet et al, 2024; Li et al, 2022; McIntyre et al, 2020; Park et al, 2021; Tan et al, 2019; Yang et al, 2023). Whether these two transcriptionally distinct clusters have different biological functions has not been investigated. Addressing this question, the current study provides evidence that peripheral CD27$^+$ IFNγ-producing γδ T cells consist of an immature Ly6C$^-$ population that converts into a mature Ly6C$^+$ population with cancer cell killing capacity. The transcriptomes of these two subsets were highly similar to that of human immature and mature γδ T cells, revealing conserved biology between species. We identify IL-27 as a phenotypic and functional regulator of mature mouse CD27$^+$Ly6C$^+$ γδ T cells and human Vγ9 Vδ2$^+$ cells. Hence, CD27$^+$ IFNγ-producing γδ T cells exist in a hierarchical state where immature cells differentiate into mature, cytotoxic cells.

# Results

## Lung CD27$^+$ γδ T cells cluster into three major groups

To better understand the heterogeneity of CD27$^+$ γδ T cells from lungs of naive mice, we refined our analysis of our scRNAseq dataset by computationally separating all *Cd27*-expressing cells from cells lacking *Cd27* expression (Edwards et al, 2023). This approach yielded 458 γδ T cells from a total of 3796. t-Distributed Stochastic Neighbor Embedding (t-SNE) was utilized for visualization of the data, which identified three distinct clusters of cells, with Cluster 2 being the most transcriptionally different from Clusters 0 and 1 (Fig. 1A). The top differentially expressed genes of Cluster 2 shared a gene expression signature with Vγ6$^+$ cells (*Cd163l1, Cxcr6, Bcl2a1b, Lgals3, Tmem176a/b, S100a4*) (Fig. 1B), which we previously described (Edwards et al, 2023). Vγ6$^+$ cells normally lack expression of CD27 protein, so we focused our investigation on Clusters 0 and 1.

Cells from Cluster 0 were enriched in NK cell-associated genes (*Cd160, Ncr1, Nkg7, Klrd1, Klrc1, Gzma*), and the cytotoxic markers Cathepsin W and cytotoxic T lymphocyte-associated protein 2 complex (*Ctsw* and *Ctla2a*). Cells from Cluster 0 were also enriched for *Ccl5, Ifng*, and *Ly6c2* (Fig. 1B–D; Table 1). In Cluster 1, we found that *Malat1, Btg1, Ccr7*, and *Fau* as well as many ribosomal

proteins (*Rps1, Rps19, Rps28* and *Rps29*) are expressed to higher degree than in the cells from Cluster 0 (Fig. 1B–D; Table 1). *Ccr7* expression was unique among these genes since its expression was largely restricted to Cluster 1, whereas only subtle differences in the expression of *Malat1, Btg1*, and *Fau* were observed between Clusters 0 and 1 (Fig. 1B–D). These differences between Clusters 0 and 1 recapitulated our previous analysis of the full 3796 lung γδ T cells and data by others, where juxtaposed *Ccr7* and *Ly6c2* expression defined two clusters of CD27$^+$ γδ T cells (du Halgouet et al, 2024; Edwards et al, 2023; Li et al, 2022; McIntyre et al, 2020). Given the association of CCR7 with immature T cells (Baeyens et al, 2015), the data suggest that cells in Cluster 1 are less differentiated than cells in Cluster 0, which are enriched in cytotoxicity-associated genes.

## Mouse γδ T-cell transcriptional signatures align with human γδ T cells

The degree of homology between mouse and human γδ T cells is controversial and poorly understood. Therefore, we investigated the transcriptomic similarity between CD27$^+$ γδ T cells from mice and human γδ T cells. We generated gene signatures from the scRNAseq data shown in Fig. 1A–D for the cytotoxic group of cells in Cluster 0 and the naive-like cells in Cluster 1, consisting of 76 and 94 genes, respectively (Table 1). These gene signatures were compared to human publicly available scRNAseq data from ~10,000 cells comprising Vγ9Vδ2 cells, Vδ1 cells, CD8$^+$ T cells, CD4$^+$ T cells, NK cells, B cells, and monocytes purified from three healthy donors (Pizzolato et al, 2019) (Fig. 1E). Single-Cell Signature_Explorer methodology was used to compare mouse and human data (Pont et al, 2019). When projected across the human dataset, the gene signatures of the two murine CD27$^+$ γδ T-cell clusters corresponded to well-defined human cell clusters. The gene signature from Cluster 0 mapped onto mature, terminally differentiated Vγ9Vδ2$^+$ cells, Vδ1$^+$ cells, NK cells, and CD8$^+$ T cells (Fig. 1F). By contrast, the gene signature from Cluster 1 mapped onto naive Vδ1$^+$ cells and αβ T cells (Fig. 1F). These data underscore the high degree of similarity of γδ T cells between species.

## Ly6C defines a subset of CD27$^+$ γδ T cells with a cytotoxic phenotype

Having identified two transcriptionally distinct subsets of CD27$^+$ γδ T cells in mice, we determined whether these two subsets could be distinguished by specific protein markers in naive mice. We chose Ly6C and CCR7 for Cluster 0 and Cluster 1, respectively, because of the robust and opposing expression levels observed by scRNAseq, as well as their established cell surface expression. Flow cytometry analysis revealed that Ly6C and CCR7 expression by CD27$^+$ γδ T cells were mutually exclusive (Fig. 2A), suggesting that each marker can distinguish two subsets of CD27$^+$ γδ T cells. However, only a small proportion (5–25%) of CD27$^+$ γδ T cells expressed CCR7 across spleen, lymph nodes (LN), and lung (Fig. 2B). By contrast, Ly6C expression defined a clear, distinct population where ~30% of CD27$^+$ γδ T cells expressed Ly6C across all tissues examined (Fig. 2B). Because CCR7 expression patterns differed between tissues, we focused on Ly6C as a marker that may globally segregate CD27$^+$ γδ T cells by subset.

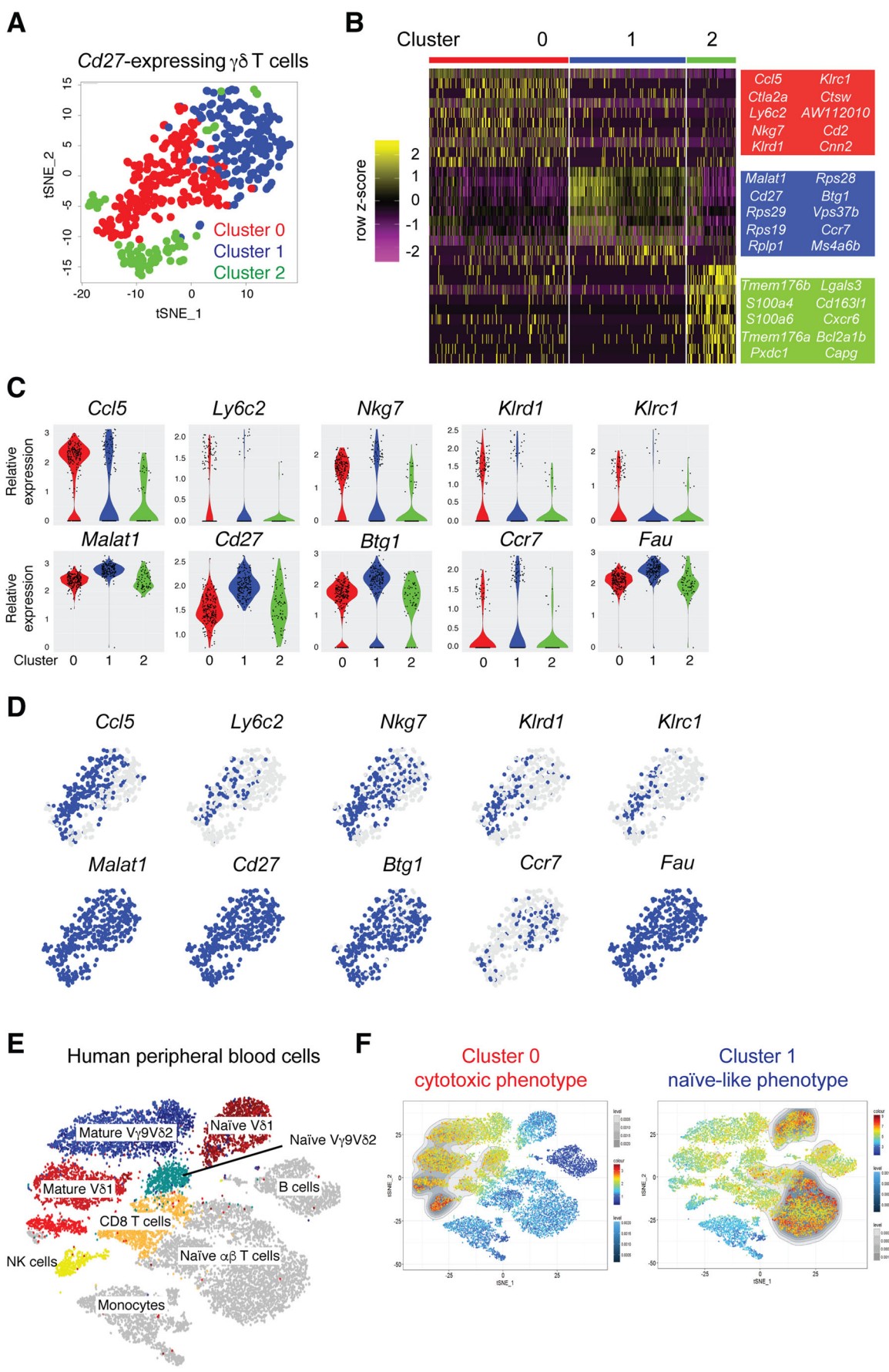

◄ **Figure 1. Mouse IFNγ-producing γδ T cells consist of two populations that are phenotypically similar to human γδ T cells.**

(A) t-SNE visualization of 458 individual lung CD27$^+$ γδ T cells color-coded by cluster. (B) Heatmap of the top ten genes from each of the three clusters identified in (A), where each column represents the gene expression profile of a single cell. Gene expression is color-coded with a scale based on z-score distribution, from low (purple) to high (yellow). (C) Violin plots showing expression levels of selected genes from the clusters identified in (A) ($n$ = 177 cells Cluster 0, 209 cells Cluster 1, 72 cells Cluster 2). (D) Feature plots of the same genes shown in (C), depicting expression levels by cell. Blue indicates high expression, and gray indicates no expression. (E) t-SNE visualization of a human scRNAseq dataset containing $2 \times 10^4$ PBMCs from three individual healthy donors (Pizzolato et al, 2019). Clusters are coded by different colors and labeled by cell type. (F) Gene signatures from Cluster 0 (left panel) and Cluster 1 (right panel) displayed on the t-SNE map from (E) by Single-Cell Signature Viewer. The colored scales represent the degree of transcriptional similarity where red indicates high similarity and dark blue indicates low similarity. The grayscale represents the density distribution of similarity scores.

The scRNAseq data indicated that CD27$^+$ γδ T cells expressing Ly6C are phenotypically different than cells lacking Ly6C expression (Fig. 1A–D). To validate this observation at the protein level, we chose four molecules from the gene signature list of Cluster 0 (Table 1) for which antibodies were available, including CD160, NKG2A (encoded by *Klrc1*), NKp46 (encoded by *Ncr1*) and IFNγ. We analyzed the expression of these molecules in CD27$^+$Ly6C$^-$ and CD27$^+$Ly6C$^+$ γδ T cells from the spleen, LN and lung. We observed that CD27$^+$Ly6C$^+$ γδ T cells displayed higher expression levels of all four molecules regardless of tissue type, when compared with CD27$^+$Ly6C$^-$ γδ T cells (Figs. 2C and EV1A). T-bet is a transcription factor that regulates *Ifng* expression in CD27$^+$ γδ T cells (Barros-Martins et al, 2016). We used T-bet reporter mice (Kadekar et al, 2020) to determine whether T-bet expression was localized to CD27$^+$Ly6C$^-$ or CD27$^+$Ly6C$^+$ γδ T cells. We found that the vast majority of CD27$^+$Ly6C$^+$ γδ T cells expressed T-bet, whereas only a minority of CD27$^+$Ly6C$^-$ γδ T cells expressed T-bet (Fig. 2D). In addition, CD27$^+$Ly6C$^+$ γδ T cells expressed higher levels of CD44 than CD27$^+$Ly6C$^-$ γδ T cells (Figs. 2E and EV1A), reminiscent of antigen-experienced CD4$^+$ and CD8$^+$ T cells. Overall, this phenotypic validation of scRNAseq data indicated that CD27$^+$Ly6C$^+$ γδ T cells (Cluster 0) represent a subset of cells with a cytotoxic phenotype, whereas CD27$^+$Ly6C$^-$ γδ T cells (Cluster 1) resemble cells in a less activated or immature state.

Given the hierarchical relationship between CD27$^+$Ly6C$^-$ and CD27$^+$Ly6C$^+$ γδ T cells, we asked when these cells appear during post-natal development. We found that CD27$^+$Ly6C$^+$ γδ T cells are measurable at low levels as early as 3 days after birth in the spleen and lung. However, these cells failed to reach maximal levels until after puberty (day 49) into adulthood (day 84 + ) (Fig. 2F).

## CD27$^+$Ly6C$^-$ and CD27$^+$Ly6C$^+$ cell subsets are present in TCR diverse populations

Ly6C expression was examined in different TCR-defined subsets of CD27$^+$ γδ T cells. In secondary lymphoid organs, CD27$^+$Vγ4$^+$ cells expressed higher levels of Ly6C than CD27$^+$Vγ1$^+$ or CD27$^+$Vγ1$^-$Vγ4$^-$ cells, but in the lung, CD27$^+$Vγ1$^+$, CD27$^+$Vγ4$^+$, and CD27$^+$Vγ1$^-$Vγ4$^-$ cells all expressed the same levels of Ly6C (Figs. 3A and EV1B). We then investigated TCR diversity within CD27$^+$Ly6C$^-$ and CD27$^+$Ly6C$^+$ γδ T cells. Across all tissues, the Vγ1 TCR was dominant among both CD27$^+$Ly6C$^-$ and CD27$^+$Ly6C$^+$ γδ T cells, comprising ~60% of total cells with the Vγ4 TCR and other TCRs making up 20% each of the total (Fig. 3B). However, this was explained by the fact that CD27$^+$ γδ T cells are largely made up of Vγ1$^+$ cells, regardless of anatomical location in which they reside (Fig. 3C).

We individually sorted CD27$^+$Vγ1$^+$Ly6C$^-$, CD27$^+$Vγ1$^+$Ly6C$^+$, CD27$^+$Vγ4$^+$Ly6C$^-$, and CD27$^+$Vγ4$^+$Ly6C$^+$ cells from pooled spleen and LNs, then investigated gene expression differences between these subsets to determine whether there was phenotypic diversity between TCR-defined subsets. Principle component analysis (PCA) based on average gene expression profiles revealed that CD27$^+$Vγ1$^+$ cells are different to CD27$^+$Vγ4$^+$ cells (Fig. 3D), and CD27$^+$Ly6C$^-$ cells are different to CD27$^+$Ly6C$^+$ cells (Fig. 3E). We examined the differences between these cells separately, and we found that CD27$^+$Vγ1$^+$Ly6C$^-$ cells and CD27$^+$Vγ4$^+$Ly6C$^-$ cells are more dissimilar than CD27$^+$Vγ1$^+$Ly6C$^+$ cells and CD27$^+$Vγ4$^+$Ly6C$^+$ cells. When comparing CD27$^+$Vγ1$^+$Ly6C$^-$ cells to CD27$^+$Vγ4$^+$Ly6C$^-$ cells, we found that 134 genes were differentially expressed between the subsets (Fig. 3F; Dataset EV1). By contrast, only 32 genes were differentially expressed between CD27$^+$Vγ1$^+$Ly6C$^+$ cells and CD27$^+$Vγ4$^+$Ly6C$^+$ cells (Fig. 3G; Dataset EV2). CD27$^+$Vγ1$^+$Ly6C$^-$ cells expressed lower levels of *Klf2, Lck*, and *Tcf7*, as well as higher levels of *Slamf6, Itga4, Cd160*, and *Eomes*, compared to CD27$^+$Vγ4$^+$Ly6C$^-$ cells (Fig. 3H). In agreement with our scRNAseq and protein expression analysis (Figs. 1 and 2), CD27$^+$Ly6C$^-$ cells—independent of their TCR—expressed higher levels of *S1pr1* and *Ccr7* than CD27$^+$Ly6C$^+$ cells; whereas CD27$^+$Ly6C$^+$ cells expressed higher levels of *Ifng, Klrc1, Ly6c1, Ncr1, Tbx21, Klrk1, Klrc2, Ccl5, Ly6c2*, and *Cd160* (Fig. 3H). These data show that regardless of the TCR they express, Ly6C$^-$ cells exhibit an immature phenotype and Ly6C$^+$ cells exhibit a mature, cytotoxic phenotype. These data from spleen- and LN-derived γδ T cells also confirm our scRNAseq data from lung γδ T cells, indicating that the phenotype of CD27$^+$Ly6C$^-$ and CD27$^+$Ly6C$^+$ γδ T cells is similar between visceral and secondary lymphoid organs.

## Ly6C$^+$ γδ T cells control tumor growth

To test the functional importance of CD27$^+$Ly6C$^-$ and CD27$^+$Ly6C$^+$ γδ T cells, these sorted populations were expanded ex vivo over 4 days with CD3/CD28 beads, IL-2, and IL-15 to generate enough material for in vitro assays. CD27$^+$Ly6C$^-$ γδ T cells expanded more readily than CD27$^+$Ly6C$^+$ γδ T cells (Fig. 4A). CD27$^+$Ly6C$^-$ γδ T cells increased about 15-fold, whereas CD27$^+$Ly6C$^+$ γδ T cells increased about eightfold (Fig. 4B). This difference in expansion was not due to a difference in cell division (Fig. 4C); however, we observed that CD27$^+$Ly6C$^+$ γδ T cells underwent more cell death than CD27$^+$Ly6C$^-$ γδ T cells (Fig. 4D). These data suggest that CD27$^+$Ly6C$^+$ γδ T cells may represent terminally differentiated cells with a shorter lifespan. After 4 days of expansion, CD27$^+$Ly6C$^-$ γδ T cells remained negative for Ly6C expression, and only a minority of CD27$^+$Ly6C$^+$ γδ T cells retained

**Table 1. Mouse gene signatures from Cluster 0 and Cluster 1 for overlay with human dataset.**

| Cluster 0 | Cluster 1 |
|---|---|
| Ccl5 | Malat1 |
| Klrc2 | Rps27 |
| Klrc1 | Ccr7 |
| Aw112010 | Rps29 |
| Nkg7 | Gm8369 |
| Klrd1 | S1pr1 |
| Ctla2a | Lef1 |
| Ly6c2 | Dusp10 |
| Xcl1 | Rpl37 |
| Fcer1g | Smc4 |
| Klre1 | Ms4a6b |
| Cd7 | Ifi27l2a |
| Cd27 | Dapl1 |
| Lck | Klf2 |
| Hopx | Rpl21 |
| Rpl37 | Cd27 |
| Samd3 | Tsc22d3 |
| Pglyrp1 | Rpl35a |
| Gramd3 | Btg1 |
| Gimap7 | Sox4 |
| Gm10260 | Rps28 |
| Tyrobp | Prkca |
| Ptprc | Fyb |
| Ncr1 | Rps17 |
| Gzmm | Sell |
| Arl4c | Ms4a4b |
| Sp100 | Saraf |
| Rnf138 | Ms4a6c |
| Cd28 | Dgka |
| Zfos1 | Cd28 |
| Ms4a6c | Klf3 |
| Txk | Atp11b |
| Sell | Satb1 |
| Klf3 | Peli1 |
| Hcst | Tcf7 |
| Fyb | Rps27rt |
| Plcxd2 | Cox7a2l |
| Cd160 | Plac8 |
| X1700025g04rik | Fam169b |
| Fasl | Crlf3 |
| Serpinb6b | Npc2 |
| Higd1a | Foxp1 |
| Efhd2 | Tspan13 |
| Cd2 | Tubb5 |

**Table 1.** (continued)

| Cluster 0 | Cluster 1 |
|---|---|
| Tm6sf1 | Lck |
| Sgk1 | Gramd3 |
| Il2rb | Sh2d1a |
| Hmgn1 | Pitpnc1 |
| Nr4a2 | Cytip |
| Neurl3 | Rasgrp2 |
| Gm19585 | Ablim1 |
| Tagap | Tuba1b |
| Fyn | Vgll4 |
| Sept6 | Il21r |
| Gimap4 | Cnp |
| Ptpn22 | Zc3hav1 |
| Rps18-ps3 | Arid4a |
| Ifitm10 | Slamf6 |
| Hn1 | Xrn2 |
| Sidt1 | Klf13 |
| Tra2b | Acp5 |
| Cldnd1 | Pim2 |
| Ugcg | Samhd1 |
| Ubr2 | Chd2 |
| Stmn1 | Rras2 |
| Hmgb2 | Chd3 |
| B4galt1 | Gm10282 |
| Ssh2 | Skap1 |
| Spry2 | Itk |
| Ifng | Stk4 |
| Cst7 | Ppm1h |
| Gzma | Arl5c |
| Peak1 | Gimap5 |
| Me2 | Stk38 |
| Dennd4a | Kras |
| Odc1 | Tmem71 |
| | Frat2 |
| | Tcp11l2 |
| | Ikzf1 |
| | Hmgn2 |
| | Ubac2 |
| | Cdkn1b |
| | Aw112010 |
| | Ssh2 |
| | Smchd1 |
| | Atp6v1d |
| | Pyhin1 |
| | Pura |

**Table 1.** (continued)

| Cluster 0 | Cluster 1 |
|-----------|-----------|
| | H2-T22 |
| | Grap2 |
| | Btf3l4 |
| | Srpk2 |
| | Tspan32 |
| | Ppm1b |

Ly6C expression (Figs. 4E and EV2A). It should be noted that TCR stimulation in the form of CD3/CD28 beads together with IL-2, and IL-15 failed to increase Ly6C expression, as these were included in both groups. The phenotype of the two expanded subsets became more similar after expansion: both subsets expressed equal levels of CD160, NKp46, and IFNγ, but NKG2A was higher on CD27⁺Ly6C⁺ γδ T cells (Figs. 4F and EV2B), as observed when cells were analyzed directly from mice (Fig. 2C). The cytotoxic function of expanded cells was tested in cancer cell killing assays, using three different mammary cancer cell lines: *K14-Cre;Trp53^{F/F}* (KP) cells, *K14-Cre;Brca1^{F/F};Trp53^{F/F}* (KB1P) cells, and E0771 cells. Expanded cells from FVB/n mice were matched to KP and KB1P cells, and expanded cells from C57BL/6 mice were matched to E0771 cells. We found that CD27⁺Ly6C⁺ γδ T cells induce more cancer cell death than CD27⁺Ly6C⁻ γδ T cells regardless of mouse background strain (Figs. 4G and EV2C), confirming the hypothesis that CD27⁺Ly6C⁺ γδ T cells have greater cytotoxic function.

The ability of CD27⁺Ly6C⁻ and CD27⁺Ly6C⁺ γδ T cells to control tumor growth in vivo was tested with the E0771 model. E0771 cells were injected in *Tcrd^{−/−}* mice to avoid interference by endogenous γδ T cells, and the expanded cell subsets were injected into tumor-bearing mice at four different intervals (Fig. 4H). Naive CD8⁺ T cells were administered as negative control. On day 15 post cancer cell injection, 2 days after the 2nd administration of T cells, tumor size increased to above 75% from baseline for ~80% of mice in PBS-, CD8⁺ T-cell-, and CD27⁺Ly6C⁻ γδ T-cell-treated groups. By contrast, tumor size increased to 75% from baseline in only ~33% of CD27⁺Ly6C⁺ γδ T-cell-treated mice (Fig. 4I). Over the course of the experiment, CD8⁺ T cells and CD27⁺Ly6C⁻ γδ T cells had little impact on the growth of tumors. Conversely, CD27⁺Ly6C⁺ γδ T cells were able to slow tumor growth and extend the survival of tumor-bearing mice when compared to control (Fig. 4J,K). Taken together, these data show that CD27⁺Ly6C⁺ γδ T cells are inherently superior over CD27⁺Ly6C⁻ γδ T cells at killing cancer cells.

## Tumors regulate the abundance, phenotype, and proliferative capacity of γδ T cells

We investigated the frequency and phenotype of CD27⁺Ly6C⁺ γδ T cells in tumor-bearing mice to determine whether tumor-derived factors influence these cells. In KP, KB1P, and E0771 mammary tumor models and the B16 melanoma model, CD27⁺Ly6C⁺ γδ T cells were more abundant in tumor tissue than the spleen, LN, or lung of tumor-bearing mice (Figs. 5A and EV3A). Across these tissues, C57BL/6 mice also exhibited higher proportions of CD27⁺Ly6C⁺ γδ T cells than FVB/n mice (Fig. 5A). When examining the phenotype of CD27⁺Ly6C⁻ and CD27⁺Ly6C⁺ γδ T cells in KP and KB1P tumor-bearing mice, we found that CD27⁺Ly6C⁺ γδ T cells expressed higher levels of CD160, NKG2A, IFNγ, and CD44 in spleen, LN, and lung (Fig. 5B), analogous to observations made in tumor-naive mice (Fig. 2C). However, CD27⁺Ly6C⁻ and CD27⁺Ly6C⁺ γδ T cells within the tumor microenvironment expressed the same levels of CD160, NKG2A, NKp46, and CD44, with NKG2A expression in KP tumors being an exception. Only IFNγ remained higher on CD27⁺Ly6C⁺ γδ T cells in both tumor models (Fig. 5B). The analysis showed that tumor-infiltrating CD27⁺Ly6C⁻ γδ T cells increased expression of each marker when compared to CD27⁺Ly6C⁻ γδ T cells in the spleen, LN, or lung, indicating that CD27⁺Ly6C⁻ γδ T cells are modified by tumors. We also measured PD-1 and CD69 on the subsets across tissues. Generally, PD-1 expression was higher on CD27⁺Ly6C⁻ cells than CD27⁺Ly6C⁺ γδ T cells (Fig. EV3B,C), in agreement with *Pdcd1* gene expression data (Fig. 3H). The expression of CD69 was relatively low on both subsets in visceral and secondary lymphoid organs, but dramatically increased in tumors (Fig. EV3D,E), suggesting that both subsets are activated in tumors. We measured Ki-67 expression in wild-type (WT), tumor-naive mice, and tumor-bearing KB1P mice as a surrogate marker for proliferation. In LN, we made two observations: (1) CD27⁺Ly6C⁺ cells expressed higher levels of Ki-67 than CD27⁺Ly6C⁻ cells in both WT and tumor-bearing KB1P mice; and (2) CD27⁺Ly6C⁻ and Ly6C⁺ cells from tumor-bearing KB1P mice expressed higher levels of Ki-67 than cells from WT mice (Figs. 5C and EV3F). While these data suggested that tumors induce expansion of cells in LN, we did not find a difference in absolute numbers of these cell subsets between WT and tumor-bearing KB1P mice (Fig. 5C). Finally, TCR usage by KP or KB1P tumor-infiltrating CD27⁺Ly6C⁻ and CD27⁺Ly6C⁺ γδ T cells was the same between subsets with no dominant TCRs emerging within either subset (Fig. 5D). These data stand in contrast to other organs where Vγ1⁺ cells were dominant (Fig. 3A). Taken together, these data indicate that tumors contain a higher proportion of CD27⁺Ly6C⁺ cells than healthy organs, and tumors influence the phenotype of both CD27⁺Ly6C⁻ and Ly6C⁺ cells.

## Ly6C⁻ cells convert into Ly6C⁺ cells

The increased ratio of CD27⁺Ly6C⁺ γδ T cells to CD27⁺Ly6C⁻ γδ T cells in tumors from four different models suggested two possibilities: either CD27⁺Ly6C⁺ γδ T cells are preferentially recruited to tumors or CD27⁺Ly6C⁻ γδ T cells convert into Ly6C⁺ cells. We failed to find support for the first hypothesis within the scRNAseq data, as no detectable chemokine receptors—other than *Ccr7, S1pr1*, and *Sell* (L-selectin), whose gene products regulate homing to lymph node—were differentially expressed between CD27⁺Ly6C⁻ and CD27⁺Ly6C⁺ γδ T cells (Fig. 1; Table 1). Therefore, we tested the second hypothesis. CD27⁺Ly6C⁻ and CD27⁺Ly6C⁺ γδ T cells were sorted separately from naive mice and immediately injected into *NOD;Rag1^{−/−};Il2rg^{−/−}* (NRG-SGM3) mice to investigate lymphopenia-driven expansion (Fig. 6A). After 7 days, we measured Ly6C expression on these cells. We discovered that about 70% of CD27⁺Ly6C⁻ γδ T cells acquire Ly6C expression when recovered from the spleen or lung (Fig. 6B,C). We also found that about 10% of recovered CD27⁺Ly6C⁺ γδ T cells lost or downregulated expression of Ly6C (Fig. 6B,D). We examined the

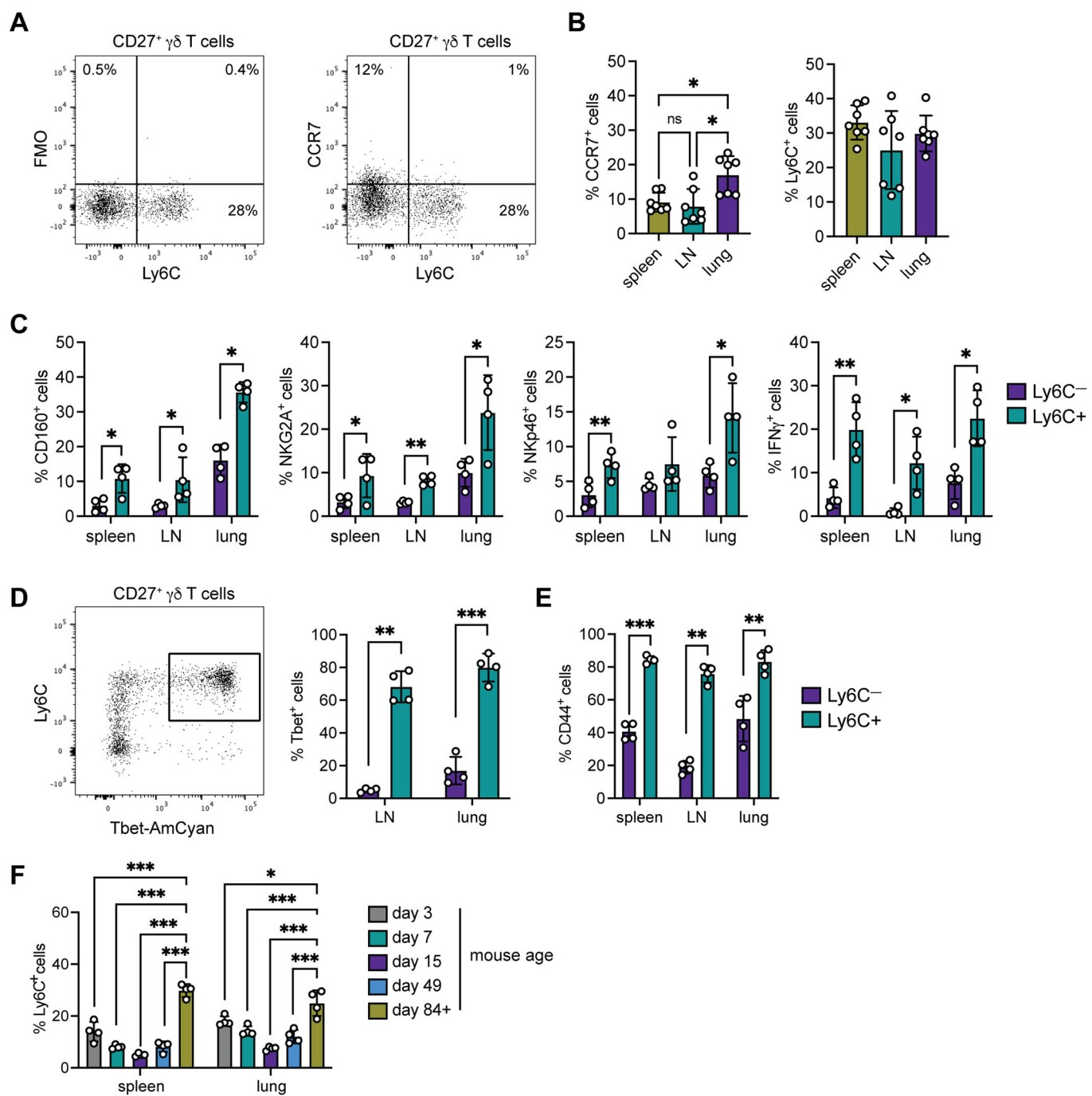

**Figure 2. Ly6C defines a subset of mature CD27⁺ γδ T cells in mice.**

(A) Dot plots of Ly6C and CCR7 staining. Viable mouse lung γδ T single cells were gated on CD3⁺ and TCRδ⁺ cells, followed by CD27⁺ cells. (B) Frequency of CCR7⁺ and Ly6C⁺ cells among CD27⁺ γδ T cells in the spleen, LN and lung (n = 7 mice/group). (C) Frequency of CD27⁺Ly6C⁻ or CD27⁺Ly6C⁺ cells expressing CD160, NKG2A, NKp46 or IFNγ in indicated tissue (n = 4 mice/group). (D) Dot plot of Ly6C and T-bet-AmCyan expression in LN after gating on CD27⁺ γδ T cells. Frequency of T-bet expression in CD27⁺Ly6C⁻ or CD27⁺Ly6C⁺ cells in LN and lung (n = 4 mice/group). (E) Frequency of CD44 expression in CD27⁺Ly6C⁻ or CD27⁺Ly6C⁺ cells (n = 4 mice/group). (F) Frequency of Ly6C⁺ cells among CD27⁺ γδ T cells in indicated tissue at various time points (n = 4 mice/group). Data information: All data are represented as mean ± SD. P values were calculated by repeated measures of one-way ANOVA followed by Tukey's posthoc test (B), paired t test (C–E), one-way ANOVA followed by Tukey's posthoc test (F). *P < 0.05, **P < 0.01, ***P < 0.001.

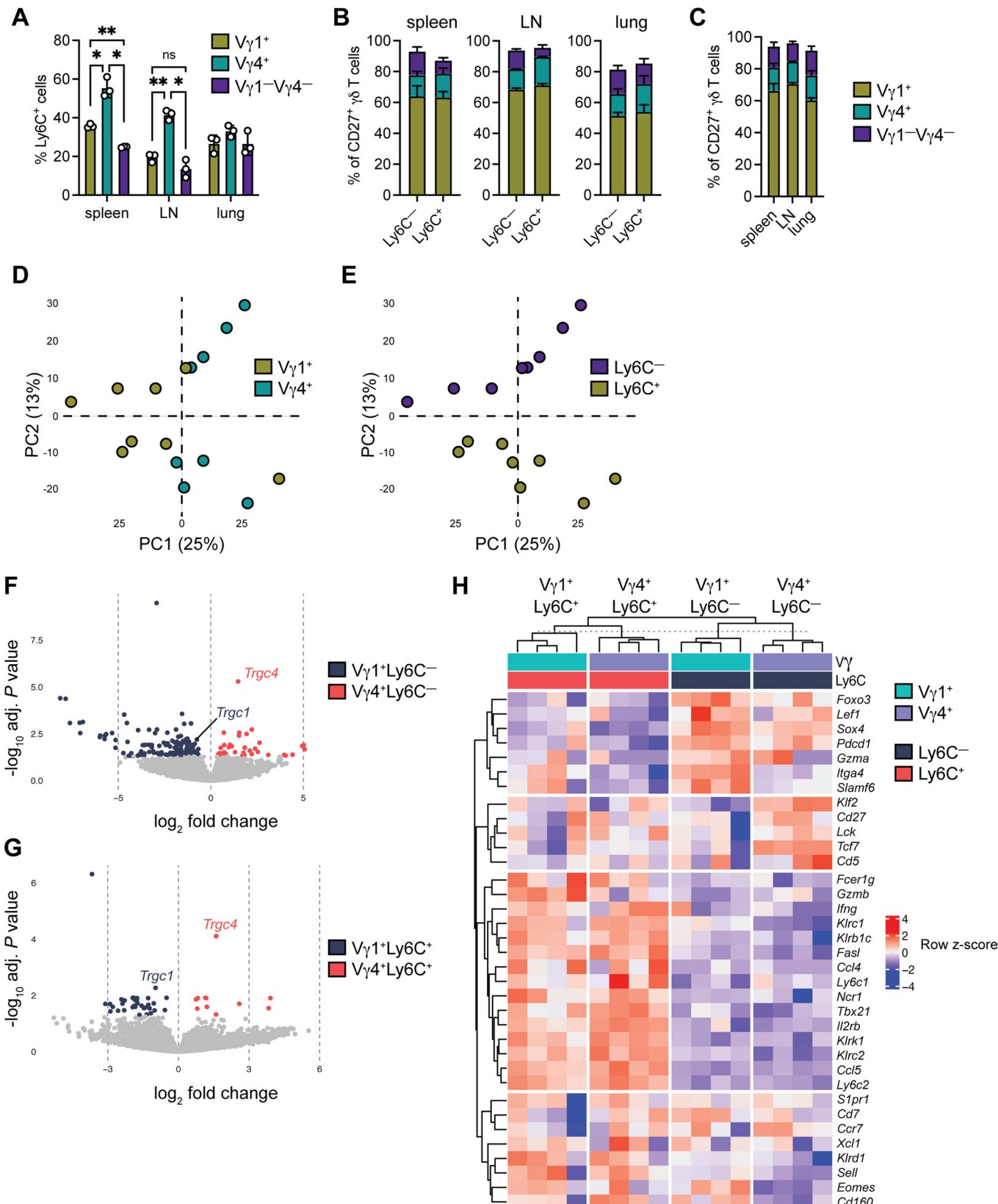

◄ **Figure 3. Distinct TCR-expressing CD27$^+$ γδ T-cell subsets exhibit phenotypic differences based on their Ly6C expression status.**

(A) Frequency of Ly6C$^+$ cells among CD27$^+$Vγ1$^+$, CD27$^+$Vγ4$^+$, and CD27$^+$Vγ1$^-$Vγ4$^-$ cells in indicated tissue ($n = 3$ female FVB/n mice/group). (B) Proportion of Vγ1$^+$, Vγ4$^+$, and Vγ1$^-$Vγ4$^-$ cells within CD27$^+$Ly6C$^-$ or CD27$^+$Ly6C$^+$ cells in indicated tissue ($n = 3$ female FVB/n mice/group). (C) The proportion of Vγ1$^+$, Vγ4$^+$, and Vγ1$^-$Vγ4$^-$ cells within total CD27$^+$ γδ T cells in indicated tissue ($n = 3$ female FVB/n mice/group). Data information: All data are represented as mean ± SD. P values were calculated by repeated measures one-way ANOVA followed by Tukey's post hoc test (A), paired $t$ test (B, C). ns not significant, *$P < 0.05$, **$P < 0.01$. (D, E) PCA analysis of RNA-seq data from sorted CD27$^+$Vγ1$^+$Ly6C$^-$, CD27$^+$Vγ1$^+$Ly6C$^+$, CD27$^+$Vγ4$^+$Ly6C$^-$, and CD27$^+$Vγ4$^+$Ly6C$^+$ cells isolated spleen and LNs of C57BL/6 mice. Each dot represents cells pooled from ten mice. Dots are colored to distinguish TCR usage in (D) or Ly6C status in (E). (F) Volcano plot representation of a comparison between the transcriptomes of CD27$^+$Vγ1$^+$Ly6C$^-$ cells and CD27$^+$Vγ4$^+$Ly6C$^-$ cells. Differentially expressed genes are denoted by blue if enriched in CD27$^+$Vγ1$^+$Ly6C$^-$ cells and, red if enriched in CD27$^+$Vγ4$^+$Ly6C$^-$ cells, whereas genes with similar expression levels are denoted in gray. Differentially expressed genes were calculated by a moderated $t$ test using the *limma* package. (G) Volcano plot representation of a comparison between the transcriptomes of CD27$^+$Vγ1$^+$Ly6C$^+$ cells and CD27$^+$Vγ4$^+$Ly6C$^+$ cells. Differentially expressed genes are denoted by blue if enriched in CD27$^+$Vγ1$^+$Ly6C$^-$ cells and, red if enriched in CD27$^+$Vγ4$^+$Ly6C$^-$ cells, whereas genes with similar expression levels are denoted in gray. Differentially expressed genes were calculated by a moderated $t$ test using the *limma* package. (H) Heatmap showing z-score normalized expression of selected genes in CD27$^+$Vγ1$^+$Ly6C$^+$, CD27$^+$Vγ4$^+$Ly6C$^+$, CD27$^+$Vγ1$^+$Ly6C$^-$, and CD27$^+$Vγ4$^+$Ly6C$^-$ cells, which is color-coded with a scale based on z-score distribution, from −4 (blue) to 4 (red).

phenotype of the recovered cells focusing on the differences between Ly6C$^-$ and Ly6C$^+$ subsets. Within the injected CD27$^+$Ly6C$^-$ γδ T-cell group, recovered Ly6C$^+$ cells expressed higher levels of CD160, NKG2A, and NKp46, but not IFNγ (Fig. 6E), which largely mirrored observations of endogenous CD27$^+$Ly6C$^-$ and CD27$^+$Ly6C$^+$ γδ T cells (Figs. 2C and 5B). In stark contrast, Ly6C$^-$ and Ly6C$^+$ subsets within the injected CD27$^+$Ly6C$^+$ γδ T-cell group maintained the same levels of CD160, NKG2A, NKp46, and IFNγ (Fig. 6F). These data corroborate but substantially extend observations by others (Lombes et al, 2015).

To further validate these findings, we performed repeated injections of CD27$^+$Ly6C$^-$ or CD27$^+$Ly6C$^+$ γδ T cells into E0771 tumor-bearing *Tcrd$^{-/-}$* mice, then analyzed these cells from tumors (Fig. 6G). Both CD27$^+$Ly6C$^-$ and CD27$^+$Ly6C$^+$ γδ T cells were recruited to tumors. As before, about 70% of CD27$^+$Ly6C$^-$ γδ T cells converted into Ly6C$^+$ cells in tumors (Fig. 6H), while about 10% of CD27$^+$Ly6C$^+$ γδ T cells lost Ly6C expression in tumors (Fig. 6H). In addition, we performed qPCR analysis of telomere length from sorted CD27$^+$Ly6C$^-$ and CD27$^+$Ly6C$^+$ γδ T cells as indication of their age. We found that CD27$^+$Ly6C$^-$ γδ T cells have longer telomeres than CD27$^+$Ly6C$^+$ γδ T cells (Fig. 6I), which suggests that CD27$^+$Ly6C$^+$ γδ T cells have undergone a higher number of cell divisions. We concluded from these collective data that CD27$^+$Ly6C$^-$ γδ T cells represent a precursor population to CD27$^+$Ly6C$^+$ γδ T cells, which represent a more terminally differentiated subset.

## IL-27 regulates CD27$^+$Ly6C$^+$ γδ T cells

Since TCR stimulation failed to upregulate Ly6C expression during the expansion of CD27$^+$Ly6C$^-$ γδ T cells (Fig. 4E), we explored whether cytokines induce Ly6C expression. We isolated total CD27$^+$ γδ T cells from naive mice and treated them with Th1-related cytokines or IL-2 family members or TGFβ. We found that IL-7, IL-12, and IL-27 increased the frequency of Ly6C-expressing cells and TGFβ downregulated Ly6C expression, whereas IL-2, IL-15, IL-18, IL-21, IFNβ, and IFNγ failed to modulate the proportion of Ly6C-expressing cells (Fig. 7A). IL-27 controls Ly6C expression on CD4$^+$ and CD8$^+$ T cells (DeLong et al, 2018), so we chose to investigate the effects of this cytokine on γδ T cells in more detail. We sorted CD27$^+$Ly6C$^-$ and CD27$^+$Ly6C$^+$ γδ T cells from naive mice and treated them with IL-27. Examination of cultured CD27$^+$Ly6C$^-$ γδ T cells revealed that about 1% of the population

expressed Ly6C, and IL-27 increased the frequency of Ly6C-expressing cells by about 3-fold as well as the MFI of Ly6C expression (Figs. 7B and EV4A,B). These data may be explained by potential contamination from false negatives in the sorted CD27$^+$Ly6C$^-$ γδ T cells, for example, CD27$^+$Ly6C$^+$ γδ T cells without Ly6C expression, since we observed that some CD27$^+$Ly6C$^+$ γδ T cells lose Ly6C expression in vivo (Fig. 6D,H). In line with Fig. 4E, in vitro cultured CD27$^+$Ly6C$^+$ γδ T cells lost expression of Ly6C; however, IL-27 increased Ly6C levels by about threefold as well as MFI of Ly6C expression (Figs. 7B and EV4A,B). We asked whether IL-27 impacted γδ T-cell proliferation during ex vivo expansion. When comparing cell numbers after 4 days in culture, IL-27 inhibited the expansion of sorted CD27$^+$Ly6C$^-$ γδ T cells, while increasing the expansion of sorted CD27$^+$Ly6C$^+$ γδ T cells (Fig. 7C), indicating that IL-27 specifically maintains CD27$^+$Ly6C$^+$ cells rather than CD27$^+$Ly6C$^-$ cells. Consequently, we asked whether CD27$^+$Ly6C$^+$ γδ T cells express higher levels of IL-27RA, making them more responsive to IL-27 stimulation. Indeed, in the spleen and LN, CD27$^+$Ly6C$^+$ γδ T cells expressed higher levels of IL-27RA than CD27$^+$Ly6C$^-$ cells, while all CD27$^+$ γδ T cells from the lung expressed the same levels of IL-27RA (Figs. 7D and EV4F).

To better understand the impact of IL-27 on CD27$^+$Ly6C$^+$ γδ T cells, we measured the frequency of these cells in WT and *Il27ra$^{-/-}$* mice. Ly6C-expressing cells were less abundant in spleen, LN, and lung from *Il27ra$^{-/-}$* mice, when compared to control (Fig. 7E). This decrease in the ratio of CD27$^+$Ly6C$^+$ γδ T cells in *Il27ra$^{-/-}$* mice was accompanied by an altered phenotype, as these cells also expressed lower levels of IFNγ, CD160, NKG2A, and NKp46 in spleen, LN, and lung (Figs. 7F and EV4C). We asked whether IL-27 signaling mediates conversion of CD27$^+$Ly6C$^-$ cells into CD27$^+$Ly6C$^+$ cells by sorting CD27$^+$Ly6C$^-$ and CD27$^+$Ly6C$^+$ γδ T cells from WT and *Il27ra$^{-/-}$* mice and injecting them immediately without in vitro culture into NRG-SGM3 mice, as described in Fig. 6. After 7 days, Ly6C levels on recovered CD27$^+$ γδ T cells were measured in spleen and lung tissue. The injected CD27$^+$Ly6C$^-$ γδ T cells from *Il27ra$^{-/-}$* mice were still capable of converting into CD27$^+$Ly6C$^+$ γδ T cells (Fig. 7G). In contrast, injected CD27$^+$Ly6C$^+$ γδ T cells from *Il27ra$^{-/-}$* mice exhibited reduced Ly6C expression when compared to injected CD27$^+$Ly6C$^+$ γδ T cells taken from WT mice (Fig. 7G). We injected E0771 mammary cancer cells into WT and *Il27ra$^{-/-}$* mice, then measured the proportion of Ly6C$^+$ cells and their phenotype in tumors. We

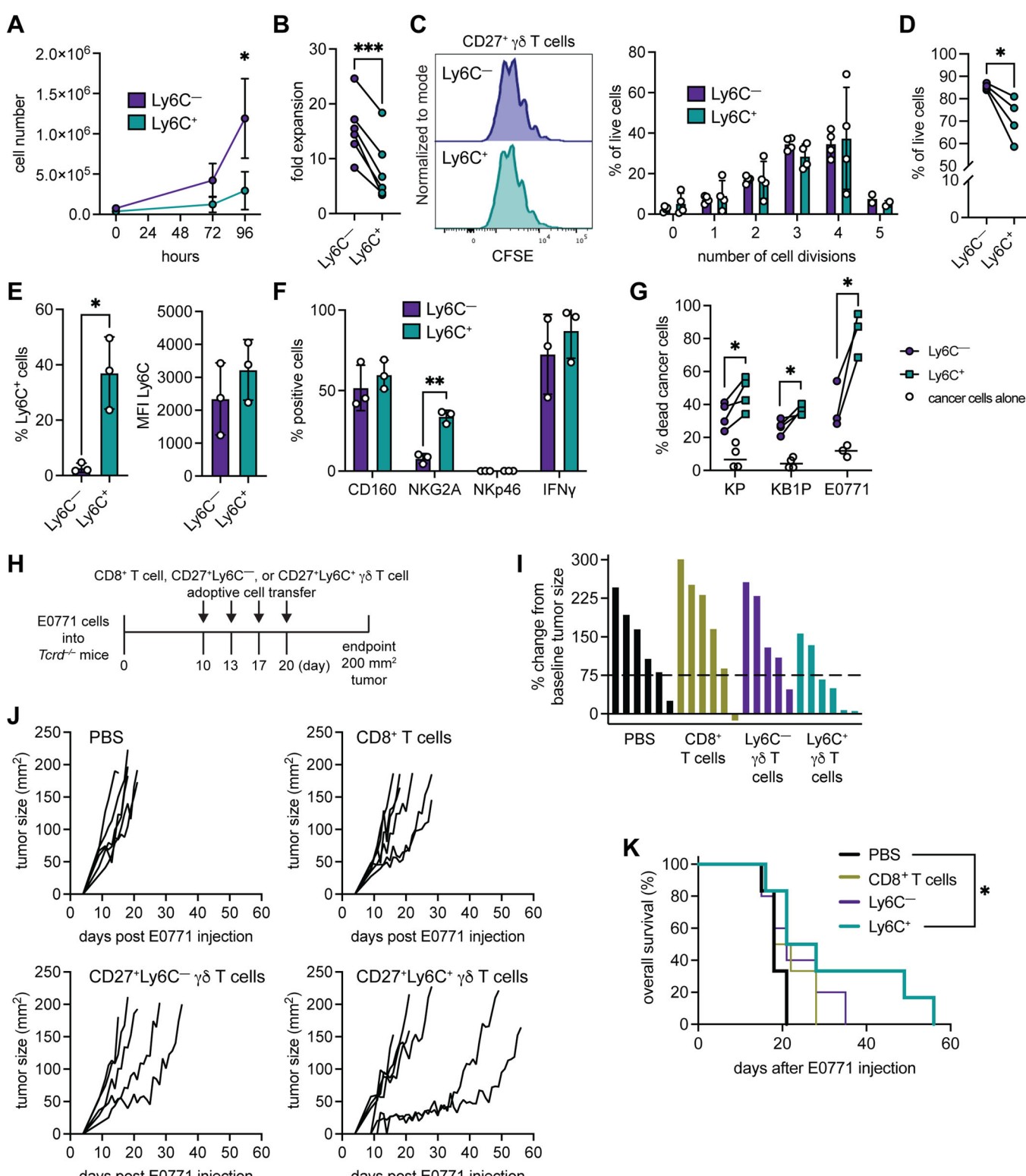

found that Ly6C-expressing cells were less frequent in tumors from *Il27ra⁻/⁻* mice (Fig. 7H). Like the observations in tumor-naive mice (Fig. 7F), tumor-infiltrating CD27⁺Ly6C⁺ γδ T cells also expressed lower levels of NKG2A, NKp46, and IFNγ (but not CD160) in *Il27ra⁻/⁻* mice (Figs. 7H and EV4D), while CD160, NKG2A,

NKp46, and IFNγ expression by tumor-infiltrating CD27⁺Ly6C⁻ γδ T cells remained the same between WT and *Il27ra⁻/⁻* mice (Figs. 7H and EV4D). These data indicate that IL-27 specifically regulates CD27⁺Ly6C⁺ γδ T-cell abundance and phenotype in steady state and cancer without influencing CD27⁺Ly6C⁻ γδ

◄

**Figure 4.** CD27+Ly6C+ γδ T cells are functionally superior at killing cancer cells than CD27+Ly6C− γδ T cells.

(A) Growth of individually sorted Ly6C− and Ly6C+ cells over 4 days. Each dot is mean of 3 biological replicates from pooled LNs and spleens of 6 FVB/n WT mice per biological replicate. (B) Fold change in expansion of Ly6C− and Ly6C+ cells over 4 days ($n = 6$ cultures of pooled cells from LNs and spleens of 6 mice). (C) Proliferation of Ly6C− and Ly6C+ cells over 4 days as determined by CFSE staining ($n = 4$ cultures of pooled cells from LNs and spleens of 6 mice). (D) Frequency of live cells in expanded Ly6C− and Ly6C+ cells over 4 days ($n = 4$ cultures of pooled cells from LNs and spleens of 6 mice). (E) Proportion and median fluorescence intensity (MFI) of Ly6C in expanded Ly6C− and Ly6C+ cells over 4 days ($n = 3$ cultures of pooled cells from LNs and spleens of 6 mice). (F) Frequency of CD160, NKG2A, NKp46 or IFNγ expression in expanded Ly6C− and Ly6C+ cells over 4 days ($n = 3$ cultures of pooled cells from LNs and spleens of 6 mice). (G) Proportion of dead KP ($n = 4$), KB1P ($n = 4$), or E0771 ($n = 3$) mammary cancer cells after co-culture with expanded Ly6C− and Ly6C+ cells for 24 h at an effector:target ratio of 10:1. Data information: All data are represented as mean ± SD. $P$ values were calculated by two-way repeated measures ANOVA (A) and paired $t$ test (B, D–G). *$P < 0.05$, **$P < 0.01$, ***$P < 0.001$. (H) Schematic of experimental setup where expanded naive, splenic CD8+ T cells, CD27+Ly6C− γδ T cells, or CD27+Ly6C+ γδ T cells were adoptively transferred into E0771-bearing $Tcrd^{-/-}$ mice on 4 separate occasions. The experiment was terminated when tumors reached 200 mm². (I) Waterfall plot of percentage change in tumor volume from tumor size at day 10 post cancer cell injection to day 15, after mice had received two injections of indicated T cells ($n = 6$ PBS, CD8, Ly6C+, $n = 5$ Ly6C−). (J) Tumor growth curves for each group ($n = 6$ PBS, CD8, Ly6C+, $n = 5$ Ly6C−). (K) Kaplan–Meier survival analysis of tumor-bearing mice that received expanded T cells ($n = 6$ PBS, CD8, Ly6C+, $n = 5$ Ly6C−). *$P < 0.05$ (log-rank test).

T cells. Ultimately, we made two conclusions from these in vitro and in vivo data: (1) IL-27 maintains Ly6C expression in CD27+Ly6C+ γδ T cells rather than mediating their conversion from CD27+Ly6C− γδ T cells, and (2) IL-27 supports cell survival and proliferation of CD27+Ly6C+ γδ T cells.

IL-27 signals through STAT1 in CD4+ and CD8+ T cells to induce Ly6C expression (DeLong et al, 2018). In agreement with these data, IL-27 treatment of sorted CD27+Ly6C− and CD27+Ly6C+ γδ T cells increased phosphorylation of STAT1, but not STAT3 or STAT4, in both subsets (Figs. 7I and EV4E). These data suggest that while both populations express the IL-27 receptor and both have the ability to respond to IL-27 stimulation, only STAT1 signaling in CD27+Ly6C+ γδ T cells is necessary for Ly6C expression and maintaining their mature phenotype.

We tested the functional consequence of IL-27 treatment on CD27+Ly6C+ γδ T cells in cancer cell line-killing assays. Sorted subsets treated with or without IL-27 were incubated with three different mammary cancer cell lines. This experiment revealed that IL-27 fails to impact the killing capacity of CD27+Ly6C− γδ T cells (Fig. 7J). For CD27+Ly6C+ γδ T-cell-cancer cell co-cultures, we obtained mixed results for the three cell lines. KP cell death was the same between the control and IL-27-treated groups. IL-27-treated Ly6C+ cells induced more cell death of KB1P cells than control. By contrast, IL-27-treated Ly6C+ cells were less efficient at inducing cell death of E0771 cells than control (Fig. 7J). The reasons for these incongruent results are unclear. Nevertheless, the altered behavior of IL-27-treated Ly6C+ cells indicates that IL-27 influences the function of this specific subset.

## IL-27 regulates human Vδ2+ cells

We investigated whether human γδ T cells are also responsive to IL-27. Initially, we tested whether γδ T-cell subsets from PBMCs from three different healthy donors have differential responses in their growth when cultured in the absence or presence of IL-27. Regardless of IL-27 treatment, both Vδ1 and Vδ2 γδ T-cell subsets expanded readily in culture (Fig. EV5A,B). PBMCs from four different healthy donors were incubated with plate-bound anti-TCRγδ and IL-27, afterwards we gated on Vδ1+ cells or Vδ2+ cells to measure their specific phenotypic changes. IL-27 failed to change the expression of CD107a, GZMB, IFNγ, NKG2A, or TNF in Vδ1+ cells. However, Vδ2+ cells upregulated CD107a and IFNγ expression levels after IL-27 treatment when compared to the

control. All other molecules remained unchanged (Figs. 8 and EV5C). These data indicate that IL-27 can specifically modulate Vδ2+ cell phenotype, uncovering conserved responsiveness to IL-27 across species.

## Discussion

Although the role of CD27+ IFNγ-producing γδ T cells in infection and cancer is well established (Ribot et al, 2021; Silva-Santos et al, 2019), the biology of these cells during disease and their adaptation to foreign insult remain poorly elucidated. This study now provides evidence that CD27+Ly6C+ γδ T cells with a cytotoxic phenotype are derived from a population of immature CD27+Ly6C− γδ T cells, both of which are found in secondary lymphoid organs and visceral organs. This transition from immature to mature γδ T cells occurs for all CD27+ IFNγ-producing γδ T cells independent of TCR usage, including Vγ1+, Vγ4+, and Vγ1−Vγ4− cells. Once differentiated, CD27+Ly6C+ γδ T cells are maintained by IL-27, which signals through the STAT1 pathway, supports their proliferation, and regulates their cytotoxic phenotype.

The acquisition of Ly6C by mature CD27+ γδ T cells mirrors αβ T-cell biology in mice, as CD4+ and CD8+ T cells upregulate Ly6C in response to antigen (Marshall et al, 2011; Walunas et al, 1995). We found that IL-27 is a cytokine that supports Ly6C expression and IFNγ, which taken together with findings by others indicate that IL-27 sustains all T-cell subsets (DeLong et al, 2018). The function of Ly6C as a glycophosphatidylinositol (GPI)-anchored membrane glycoprotein is poorly defined. Ly6C expressed by central memory CD8+ T cells is involved in adhesion to endothelial cells and homing to LN (Hanninen et al, 1997; Hanninen et al, 2011), so this molecule may also help to direct CD27+Ly6C+ γδ T cell to specific locations.

The stimulus for conversion of Ly6C− γδ T cells into Ly6C+ γδ T cells remains undefined. Unlike αβ T cells, γδTCRs bind host-encoded ligands, such as members of the butyrophilin family, endothelial protein C receptor (EPCR), or other MHC-like molecules, rather than MHC-presented peptides (Willcox and Willcox, 2019). Therefore, stimulation of a γδTCR through a stress-induced ligand, such as EPCR, which occurs in viral infection (Mantri and St John, 2019), may differentiate CD27+Ly6C− γδ T cells into CD27+Ly6C+ γδ T cells. However, this ligand (or ligands) would have to cross-react with several γδTCRs, since we

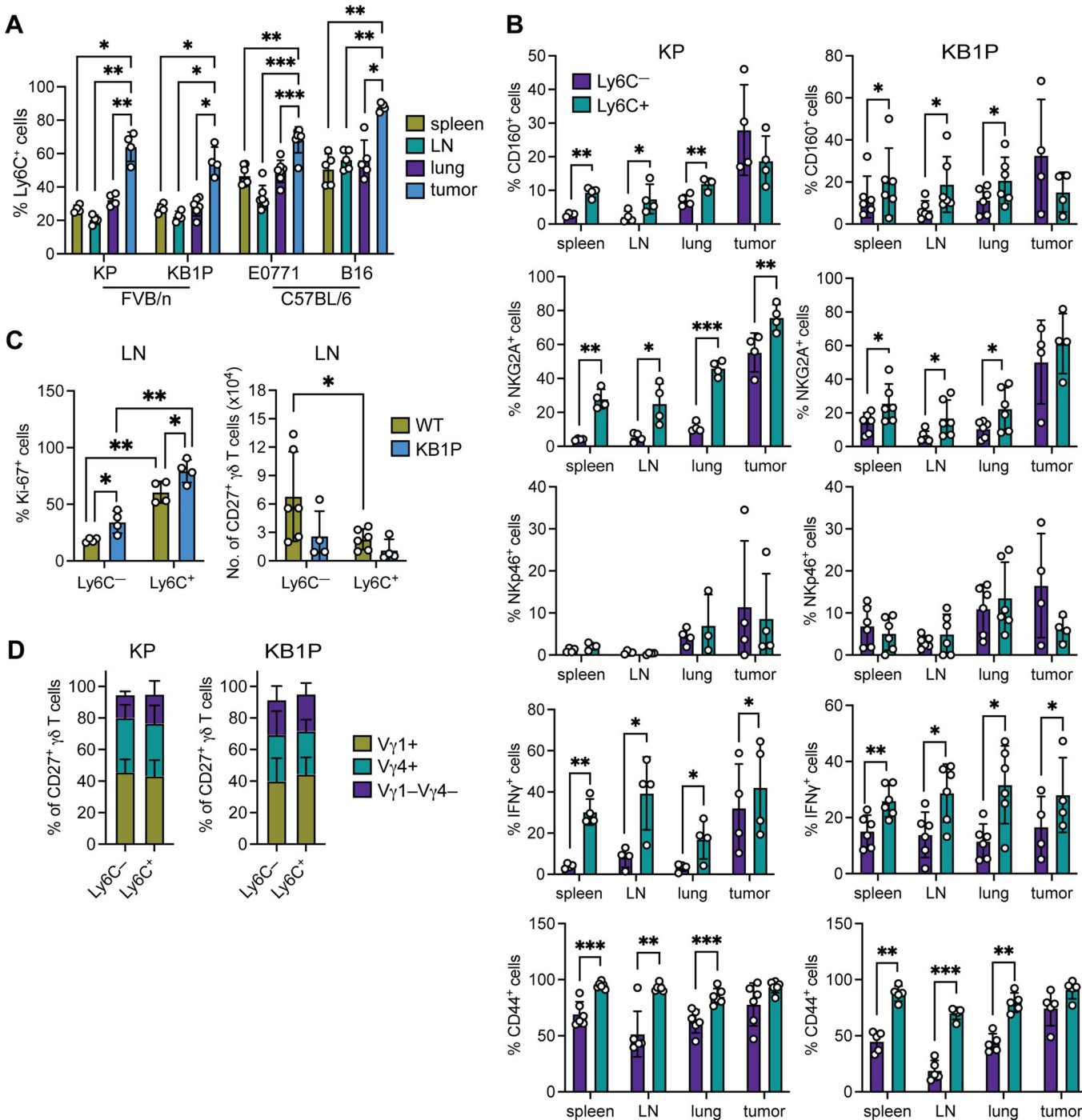

**Figure 5. CD27⁺Ly6C⁺ γδ T cells are enriched in tumors.**

(A) Frequency of Ly6C⁺ cells among CD27⁺ γδ T cells in the spleen, LN (pooled axillary, brachial and inguinal representing draining and non-draining tissue), lung and tumor of indicated tumor model (*n* = 4–7 mice/group). (B) Frequency of CD27⁺Ly6C⁻ or CD27⁺Ly6C⁺ cells expressing CD160, NKG2A, NKp46, IFNγ or CD44 in indicated tissue from KP or KB1P tumor models (*n* = 4 KP, 4–6 KB1P mice/group). (C) Frequency and absolute numbers of CD27⁺Ly6C⁻ or CD27⁺Ly6C⁺ cells expressing Ki-67 in LN of WT (*n* = 4 mice/group) or KB1P tumor-bearing (*n* = 4 mice/group) mice. *$P < 0.05$, **$P < 0.01$ (paired and unpaired *t* test). (D) Proportion of Vγ1⁺, Vγ4⁺, and Vγ1⁻Vγ4⁻ cells within CD27⁺Ly6C⁻ or CD27⁺Ly6C⁺ cells in KP or KB1P tumor tissue (*n* = 6 KP, 5 KB1P mice/group). Data information: All data are represented as mean ± SD. *P* values were calculated by repeated measures one-way ANOVA followed by Tukey's posthoc test (A), paired *t* test (B–D), and unpaired *t* test (C). *$P < 0.05$, **$P < 0.01$, ***$P < 0.001$.

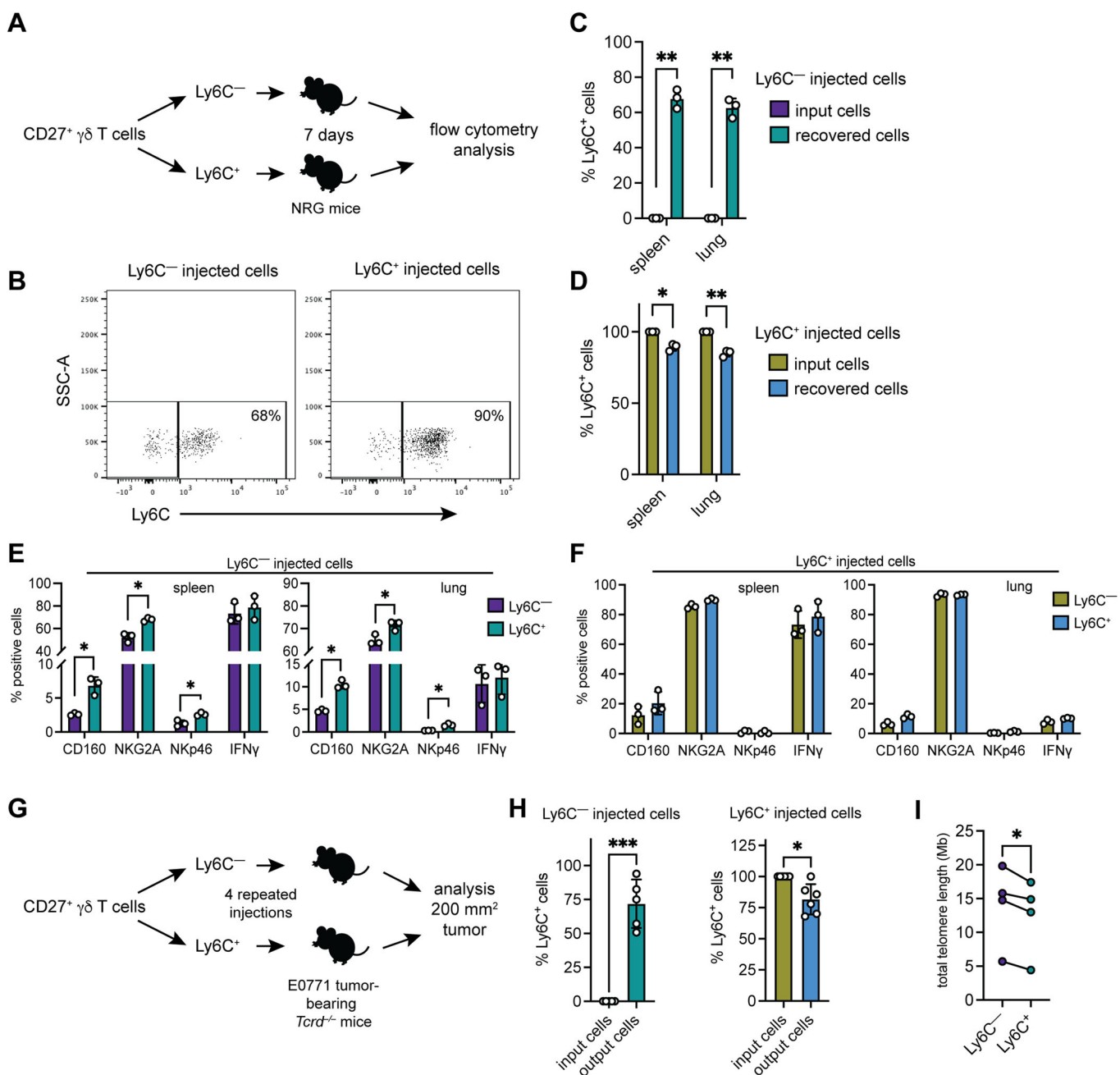

**Figure 6.  CD27+Ly6C− γδ T cells convert into CD27+Ly6C+ γδ T cells.**

(A) Schematic design of experimental method where sorted Ly6C− and Ly6C+ γδ T cells were directly injected into NRG-SGM3 mice and were then analyzed after 7 days. (B) Dot plots of Ly6C expression in recovered CD27+ γδ T cells from spleens of NRG-SGM3 mice. (C, D) Frequency of Ly6C+ cells within input cell populations and recovered cells in indicated tissue of NRG-SGM3 mice (n = 3 mice/group). (E, F) Phenotypic analysis of recovered Ly6C− and Ly6C+ cells 7 days after injection of Ly6C− and Ly6C+ cells into NRG-SGM3 mice showing the frequency of CD160, NKG2A, NKp46 or IFNγ expression in indicated tissue (n = 3 mice/group). (G) Schematic design of experimental method where sorted and expanded Ly6C− and Ly6C+ cells were injected into E0771-bearing *Tcrd−/−* mice at 4 repeated intervals then analyzed when tumors reached 200 mm². (H) Frequency of Ly6C+ cells within input cell populations and recovered cells in E0771 tumor tissue from *Tcrd−/−* mice (n = 5 mice/Ly6C− group, 6 mice/Ly6C+ group). (I) Total telomere length (per diploid cell in megabases (Mb)) as determined by qPCR of genomic DNA from sorted CD27+Ly6C− and CD27+Ly6C+ cells (n = 4 cultured cell pools from LN and spleen of 6 C57BL/6 female mice). Data information: All data are represented as mean ± SD. P values were calculated by paired t test (C–F, H, I). *P < 0.05, **P < 0.01, ***P < 0.001.

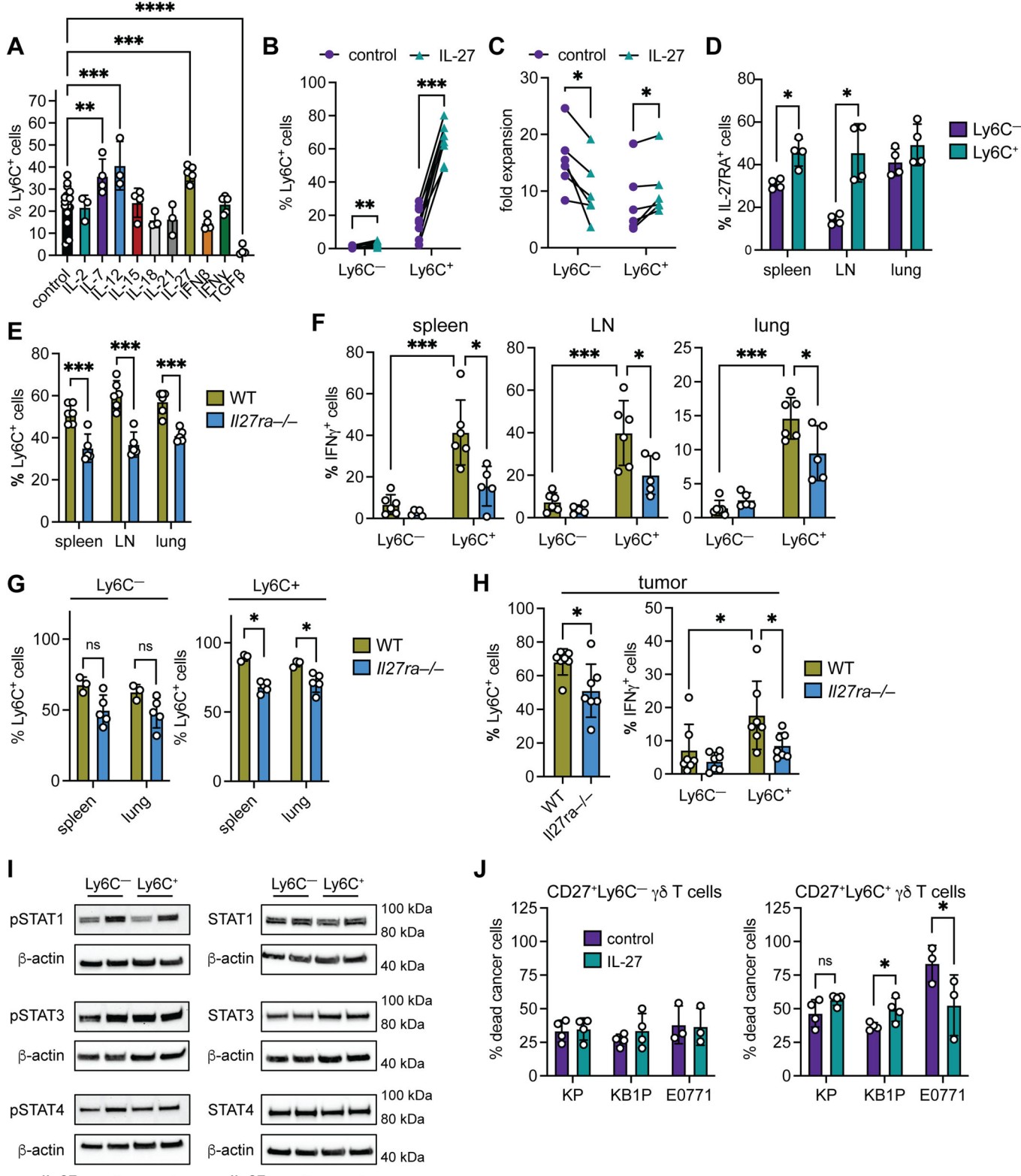

**Figure 7.  IL-27 primes CD27⁺Ly6C⁺ γδ T cells.**

(A) Frequency of Ly6C⁺ cells in cultured CD27⁺ γδ T cells after 4 days with CD3/CD28 beads (control) and indicated cytokine stimulation (control $n = 20$, stimulated $n = 3$–5 cultured cells from pooled LNs and spleens of 2 mice). (B) Frequency of Ly6C⁺ cells in sorted Ly6C⁻ and Ly6C⁺ cells treated with CD3/CD28 beads, IL-2, and IL-15 (control), with IL-27 as indicated ($n = 8$ expanded cells from pooled LNs and spleens of 6 mice). (C) Fold change in expansion of sorted Ly6C⁻ and Ly6C⁺ cells over 4 days treated as indicated ($n = 6$ replicates from pooled cells from 6 mice). (D) Frequency of IL-27RA⁺ cells among CD27⁺Ly6C⁻ and CD27⁺Ly6C⁺ γδ T cells analyzed by flow cytometry from indicated tissues of FVB/n WT mice ($n = 4$ mice/group). (E) Frequency of Ly6C⁺ cells in indicated tissues from C57BL/6 WT ($n = 6$) and *Il27ra⁻ᐟ⁻* ($n = 5$) mice. (F) Proportion of IFNγ-expressing Ly6C⁻ and Ly6C⁺ cells in indicated tissues from C57BL/6 WT ($n = 6$) or *Il27ra⁻ᐟ⁻* ($n = 5$) mice. (G) Frequency of Ly6C⁺ cells within Ly6C⁻ and Ly6C⁺ input cell populations from C57BL/6 WT or *Il27ra⁻ᐟ⁻* mice and recovered cells in indicated tissue of NRG-SGM3 mice ($n = 3$ mice/WT group, $n = 5$ mice/*Il27ra⁻ᐟ⁻* group). (H) Frequency of Ly6C⁺ cells in E0771 tumor tissue from C57BL/6 WT or *Il27ra⁻ᐟ⁻* mice. The proportion of IFNγ-expressing Ly6C⁻ and Ly6C⁺ cells in tumor tissue from C57BL/6 WT or *Il27ra⁻ᐟ⁻* mice ($n = 7$ mice/group). (I) Western blot analysis of indicated proteins in sorted Ly6C⁻ and Ly6C⁺ cells treated with or without IL-27. Actin was used as a loading control, and each corresponding loading control is depicted under the respective blot. (J) Proportion of dead KP ($n = 4$), KB1P ($n = 4$), or E0771 ($n = 3$) mammary cancer cells after 24 h co-culture with expanded Ly6C⁻ and Ly6C⁺ treated with IL-27 as indicated. Data information: All data are represented as mean ± SD. $P$ values were calculated by one-way ANOVA followed by Tukey's posthoc test (A), paired $t$ test (B–D, F, H, J), unpaired $t$ test (E–H). *$P < 0.05$, **$P < 0.01$, ***$P < 0.001$, ****$P < 0.0001$. Source data are available online for this figure.

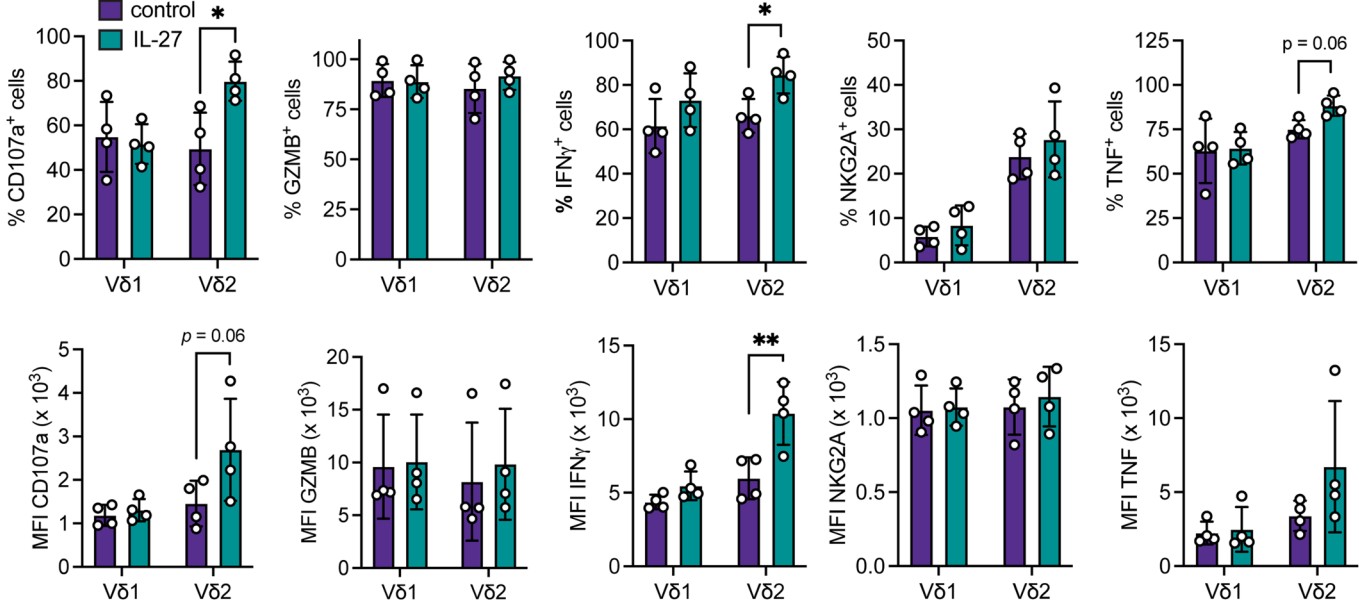

**Figure 8.  IL-27 primes human Vδ2 γδ T cells.**

Frequency and MFI of indicated proteins in Vδ1 or Vδ2 cells treated with IL-2, and IL-15 (control), and with IL-27 as indicated ($n = 4$ cells from one healthy human PBMC donor). Data information: All data are represented as mean ± SD. $P$ values were calculated by paired $t$ test. *$P < 0.05$, **$P < 0.01$.

show here that all Vγ1⁺, Vγ4⁺, and Vγ1⁻Vγ4⁻ cells undergo transition from immature to mature populations. Since *Ly6c2* gene expression is regulated by T-bet (Matsuda et al, 2006), cytokines that specifically activate transcription of *Tbx21*, the gene encoding T-bet, are also conversion-inducing candidates. Some cytokines may work in concert with TCR stimulation to drive the maturation of CD27⁺ γδ T cells. Other questions still outstanding concern the location of CD27⁺ γδ T-cell differentiation and the lifespan of these cells. Where CD27⁺Ly6C⁻ γδ T cells differentiate into CD27⁺Ly6C⁺ γδ T cells and whether this occurs in secondary lymphoid organs like αβ T cells is unknown.

Although γδ T cells are generally considered innate-like, increasing evidence supports the concept that Vδ1 and Vδ2 subsets of human γδ T cells undergo adaptive-like biology where naive-like cells transition to mature cells and clonally expand (Davey et al, 2018; Davey et al, 2017; Hunter et al, 2018; McMurray et al, 2022; Ravens et al, 2017). This conversion can be induced by

viral or parasitic infection; however, whether TCR antigens mediate this conversion is unknown. We found that the gene expression signatures of CD27⁺Ly6C⁺ and CD27⁺Ly6C⁻ γδ T cells directly correspond to the differing transcriptomes of human mature and immature γδ T cells, respectively, providing evidence of conserved homology between mouse and human γδ T cells. This in combination with the superior anti-tumor function of CD27⁺Ly6C⁺ γδ T cells provides a rationale to use mice for the study of anti-tumorigenic human γδ T cells. Given the increased interest in exploiting Vδ1⁺ and Vγ9Vδ2⁺ cells for cancer immunotherapy (Mensurado et al, 2023; Sebestyen et al, 2020; Silva-Santos et al, 2019), the use of mouse cells as surrogates for human cells in syngeneic cancer models should overcome several limitations in the field, such as the influence of γδ T-cell products on other immune cells. A fully syngeneic platform has considerable potential to improve γδ T-cell-based immunotherapies for cancer patients.

# Methods

## Reagents and tools table

| Reagent/resource | Reference or source | Identifier or catalog number |
|---|---|---|
| **Antibodies** | | |
| CCR7 (CD197), PE-Dazzle594, 4B12, 1:400 | BioLegend | Cat# 120121 |
| CD3, BV650, 17A2, 1:100 | BioLegend | Cat# 100229 |
| CD3 (human), APC-Cy7, OKT3, 1:100 | BioLegend | Cat# 317341 |
| CD3ε, FITC, 145-2C11, 1:100 | eBioscience | Cat# 100305 |
| CD4, BV605, RM4-5, 1:100 | BioLegend | Cat# 100547 |
| CD4, APC-eFluor780, GK1.5, 1:100 | Invitrogen | Cat# 47-0041-80 |
| CD8, BUV395, 53-6.7, 1:100 | BD Biosciences | Cat# 563786 |
| CD8, BUV805, 53-6.7, 1:200 | BD Biosciences | Cat# 612898 |
| CD8, APC-eFluor780, 53-6.7, 1:100 | Invitrogen | Cat# 47-0081-82 |
| CD11b, APC-eFluor780, M1/70, 1:800 | eBioscience | Cat# 47-0112-80 |
| CD11b, BV785, M1/70, 1:800 | BioLegend | Cat# 101243 |
| CD11c, APC-eFluor780, N418, 1:200 | Invitrogen | Cat# 47-0114-82 |
| CD19, APC-eFluor 780, 1D3, 1:400 | eBioscience | Cat# 47-0193-82 |
| CD19, FITC, eBio1D3, 1:800 | eBioscience | Cat# 11-0193-82 |
| CD27, BV510, LG.3A10, 1:200 | BD Biosciences | Cat# 563605 |
| CD27, PE-Dazzle594, LG.3A10, 1:400 | BioLegend | Cat# 124227 |
| CD44, PerCP-Cy5.5, IM7, 1:100 | BioLegend | Cat# 103031 |
| CD44, BV510, IM7, 1:200 | BD Biosciences | Cat# 563114 |
| CD69, BV510, H1.2F3, 1:50 | BioLegend | Cat# 104531 |
| CD160, PerCP-Cy5.5, 7H1, 1:100 | BioLegend | Cat# 143007 |
| Granzyme B (mouse/human), AF647, GB11, 1:50 | BioLegend | Cat# 515405 |
| IFNγ, PE-Cy7, XMG1.2, 1:200 | eBioscience | Cat# 25-7311-41 |
| IFNγ, eFluor450, XMG1.2, 1:200 | Invitrogen | Cat# 48-7311-82 |
| IFNγ (human), BV421, B27, 1:50 | BioLegend | Cat# 506537 |
| IL-27RA, PE, W16125D, 1:100 | BioLegend | Cat# 159003 |
| Ki-67, BV421, 16A8, 1:200 | BioLegend | Cat# 652411 |
| Ly6C, AF700, HK1.4, 1:50 | BioLegend | Cat# 128023 |
| Ly6C, PE-Dazzle594, HK1.4, 1:200 | BioLegend | Cat# 128043 |
| Ly6C, PE, HK1.4, 1:200 | BioLegend | Cat# 128007 |
| NK1.1, BUV395, PK136, 1:100 | BD Biosciences | Cat# 564144 |
| NK1.1, BV421, PK136, 1:50 | BioLegend | Cat# 108731 |
| NKG2A (human), BV711, 131411, 1:50 | BD Biosciences | Cat# 747919 |
| NKG2A/C/E, BV605, 20d5, 1:200 | BD Biosciences | Cat# 564382 |
| NKG2A/C/E, BV711, 20d5, 1:50 | BD Biosciences | Cat# 740758 |
| NKp46, FITC, 29A1.4, 1:100 | BioLegend | Cat# 137605 |
| NKp46, BV421, 29A1.4, 1:100 | BioLegend | Cat# 137611 |
| PD-1, PE-Cy7, 29 F.1a12, 1:100 | BioLegend | Cat# 135215 |
| TER-119, FITC, TER-119, 1:100 | eBioscience | Cat# 11-5921-82 |

| Reagent/resource | Reference or source | Identifier or catalog number |
|---|---|---|
| TNFα (human), BV785, Mab11, 1:100 | BioLegend | Cat# 502947 |
| TCRδ, FITC, GL3, 1:200 | eBioscience | Cat# 11-5711-82 |
| TCRδ, PE, GL3, 1:100 | BioLegend | Cat# 118107 |
| TCRδ, PerCP-Cy5.5, GL3, 1:200 | BioLegend | Cat# 118117 |
| TCR Vδ1 (human), FITC, REA173, 1:50 | Miltenyi Biotec | Cat# 130-118-362 |
| TCR Vδ2 (human), PE-Cy7, B6, 1:50 | BioLegend | Cat# 331421 |
| TCR Vγ1, PE, 2.11, 1:200 | BioLegend | Cat# 141105 |
| TCR Vγ4, APC, UC3-10A6, 1:100 | BioLegend | Cat# 137707 |

## Mice

Animal experiments were carried out in line with the Animals (Scientific Procedures) Act 1986 and the EU Directive 2010 and sanctioned by the Local Ethical Review Process at the Cancer Research UK Scotland Institute and the University of Glasgow under license 70/8645 and PP6345023 to Karen Blyth and license PP0826467 to Seth Coffelt. Mice were kept in individually ventilated cages in a barrier in-house facility with a 12-h light/dark cycle and free access to food and water. Female mice were used in this study, aged 12–20 weeks except where indicated. FVB/n and C57BL/6J mice were purchased from Charles River. Timed matings for FVB/n mice were set up to obtain pups at 3, 7, 15, 49, and 84 days after birth. The *K14-Cre;Trp53^{F/F}* (KP) and *K14-Cre;Brca1^{F/F};Trp53^{F/F}* (KB1P) colonies (Liu et al, 2007) were gifted from Jos Jonkers (Netherlands Cancer Institute), and back-crossed onto the FVB/n background for 6 generations. *Tcrd^{−/−}* mice (RRID:IMSR_JAX:002120), *Tcrb^{−/−}* mice (RRID:IMSR_JAX:002118), and *Il27ra^{−/−}* mice (RRID:IMSR_JAX:018078) on C57BL/6 J background were purchased from the Jackson Laboratory. *Tbx21*-AmCyan mice were obtained and maintained at the Technical University of Denmark as described (Kadekar et al, 2020). *NOD.Cg-Rag1^{tm1Mom};Il2rg^{tm1Wjl};Tg(CMV-IL3,CSF2,KITLG)1Eav/J* (called NRG-SGM3) mice (RRID:IMSR_JAX:024099) were purchased from the Jackson Laboratory.

KP and KB1P mice were used when a mammary tumor reached >120 mm². Cre-negative, tumor-free, littermate females were used as controls. For the E0771 model, C57BL/6J mice were injected with 50 μL containing $2.5 \times 10^5$ E0771 cells in 1:1 PBS:Matrigel (356231, Corning) mix into the fourth mammary fat pad. For the B16 model, C57BL/6J mice were injected subcutaneously with either 50 μL containing $2.5 \times 10^5$ B16-F1 melanoma cells in a 1:1 PBS:Matrigel mixture or with 50 μL of vehicle PBS:Matrigel as a control. Tumors were measured three times per week using callipers by researchers blinded to the experimental groups. Once tumors reached 15 mm in any direction, mice were analyzed by flow cytometry.

## Cell lines

KP and KB1P cell lines were generated as described (Millar et al, 2020). E0771 cells were obtained from Sara Zanivan (CRUK Scotland Institute), and B16-F1 cells were obtained from Laura Machesky (CRUK Scotland Institute). KP, KB1P, and B16-F1 cells were maintained in DMEM, 10% FCS, 100 U/mL penicillin, 100 μg/

mL streptomycin, 2 mM glutamine, and 10 mM HEPES. E0771 cells were maintained in RPMI-1640 medium, 10% FCS, 100 U/mL penicillin, 100 µg/mL streptomycin, 2 mM glutamine and 10 mM HEPES. No authentication was performed. *Mycoplasma* testing was performed monthly. After thawing, cells were not used past ten passages.

## Single-cell RNA sequencing and computational analysis

scRNAseq of γδ T cells from the lungs of wild-type mice was performed as previously described (Edwards et al, 2023). After quality control for the removal of cells with less than 200 or more than 3000 genes, genes in less than 3 cells and cells with more than 10% reads from mitochondrial genes, followed by a batch correction, we isolated the cells with *Cd27* read values >0. This selection resulted in single-cell transcriptomes from 458 CD27+ γδ T cells. The first 9 components of the principal component analysis (PCA) were used for unsupervised K-nearest clustering, and non-linear dimensional reduction using t-distributed Stochastic Neighbor Embedding (t-SNE) was utilized for visualization of the data. Then the data were analyzed to determine the differentially expressed genes of clusters 0, 1 and 2. Genes from the murine cell clusters 0- and 1-defining signatures were converted to their human orthologs from the Ensembl database using the martview interface (http://www.ensembl.org/biomart/martview). Then, by using Single-Cell_Signature_Explorer (Pont et al, 2019), these signatures were scored for each single cell of an already annotated dataset of ~8k human purified γδ T cells and PBMCs (Pizzolato et al, 2019). The signature scores were visualized as heatmap projected on the dataset t-SNE, with contours around those cells scoring > superior quartile.

## RNA sequencing

RNA was extracted from sorted CD27 + Vγ1+Ly6C−, Vγ1+Ly6C+, Vγ4+Ly6C−, and Vγ4+Ly6C+ cells using Qiagen RNAeasy kit (74104) according to the manufacturer's instructions. RNA-seq was performed using an SMARTer Stranded Total RNA-Seq Kit v2 (634413 Takara), then run on an Illumina NextSeq using the High Output 75 cycles kit (2 × 36 cycles, paired-end reads, single index). Raw sequencing reads were processed using sci_wiz (Ojo et al, 2024). The raw sequence quality was assessed using the FastQC algorithm version 0.11.8. Sequences were trimmed to remove adaptor sequences and low-quality base calls, defined by a Phred score of <20, using FastP. The trimmed sequences were aligned to the mouse genome build GRCm38.98 using STAR, then raw counts per gene were determined using FeatureCounts version 1.6.4. Differential expression analysis was performed using the R package, limma (version 3.56.1) (Ritchie et al, 2015), and PCA was performed using R base functions and VST normalization via DEseq2 (Love et al, 2014). Heatmaps, volcano plots, and PCA visualization was achieved via a combination of cowplot (version 1.1.E) (Wilke, 2024), ggplot2 (Wickham, 2016), ComplexHeatmap (version 2.16.0) (Gu et al, 2016).

## Multi-parameter flow cytometry

Mouse tissues were processed for flow cytometry analysis as previously described (Edwards et al, 2023). Briefly, after 3 h of stimulation with PMA and ionomycin together with Brefeldin A when necessary, single-cell suspensions were incubated for FcR block (TruStain FcX, anti-mouse CD16/32, 101320, BioLegend) in 0.5% BSA/PBS buffer for 20 min on ice. Cell surface antibodies were added for 30 min at 4 °C in the dark. Cell surface molecules were stained, and dead cells were identified with Zombie Green or Zombie NIR viability dye (423112 or 423106, BioLegend). Intracellular staining was performed after fixation and permeabilization with a kit (00-5523-00, eBioscience). Fluorescence minus one (FMO) controls were prepared to set gating controls. Sorted γδ T cells were stained with 2.5 nM CFSE Cell Division Tracker Kit (423801, BioLegend) for 20 min at 37 °C immediately after sorting. Cancer cell death in co-culture experiments was measured by DAPI incorporation. Data were acquired on BD LSRFortessa or LSRII using DIVA acquisition software. Data analysis was performed with FlowJo software v9 and v10.8.1 (FlowJo, LLC). Fluorescently conjugated antibodies used in this study are listed in the Reagents table.

## γδ T-cell isolation, expansion, and treatment

For bulk γδ T-cell isolation, mouse TCRγ/δ+ T-cell Isolation Kit (130-092-125, Milenyi Biotec) was used according to the manufacturer's instructions. Final positive selection was performed twice in order to increase purity.

For sorting of Ly6C− and Ly6C+ γδT-cell populations, single-cell suspensions of pooled axillary and inguinal LNs and spleens were generated from FVB/n or C57BL/6J mice. Up to $1 \times 10^7$ cells were resuspended in 100 µL 1× MojoSort Buffer (480017, BioLegend) and supplemented with 10 µL of self-made biotinylated antibody depletion cocktail containing antibodies CD4, CD8, CD11b, CD19, B220, and TER-119. After incubation of 15 min at 4 °C, 10 µL per $1 \times 10^7$ cells of MojoSort Streptavidin Nanobeads (480016, BioLegend) was added and incubated for 15 min at 4 °C. Final volumes were adjusted to a total of 2.5 mL per reaction and placed into a MojoSort magnet (480019, BioLegend) for 5 min. Negatively selected, enriched γδ T-cell suspension was then stained for sorting as for flow cytometry and sorted using a BD Aria sorter. DAPI was added for live/dead exclusion just before sorting. Unstained and single-color controls were prepared for compensation and gating. Cells were sorted on single cells >live cells (DAPI−) > CD3+TCRδ+ > CD27+ > Ly6C− vs. Ly6C+. To facilitate viability, samples were sorted at 4 °C and collected in IMDM medium supplemented with 50% FCS.

Isolated γδ T-cell populations were plated at a density of $4 \times 10^4$ per well in a 96-well U-bottom plate (650180, Greiner Bio-One). Cells were incubated with IMDM medium, 10% FCS, 100 U/mL penicillin, 100 µg/mL streptomycin, 2 mM glutamine, 50 µM βME, Dynabeads Mouse T-Activator CD3/CD28 (11452D, ThermoFisher Scientific) at a 1:1 cell:bead ratio, 10 ng/mL IL-2 (212-12, PeproTech), and 10 ng/mL IL-15 (210-15, PeproTech). On day 2 of expansion, cells were split 1:2 and supplemented with fresh (cytokine) medium without Dynabeads. On day 3, cells were collected, Dynabeads were washed off using a magnet for Eppendorf tubes, counted and re-plated as $4 \times 10^4$ per well in fresh medium with respective cytokine condition and addition of fresh Dynabeads. On day 4, cells were collected, Dynabeads washed off, counted, and used for downstream assays or flow cytometry staining and analysis. Where indicated, cells also received 10 ng/mL

IL-7 (217-17, PeproTech), 50 ng/mL IL-27 (577402, BioLegend), 10 ng/mL IFNβ (581302, BioLegend), 10 ng/mL IFNγ (315-05, PeproTech), or 5 ng/mL TGFβ (100-21, PeproTech).

## Co-culture of cancer cell lines and γδ T cells

Cancer cell lines were plated at a density of 10,000 cells in 100 μL IMDM medium, 10% FCS, 100 U/mL penicillin, 100 μg/mL streptomycin, 2 mM glutamine, 50 μM βME in a flat bottom 96-well plate. After 6 h, $10^5$ expanded γδ T cells were added to wells in 100 μL of the same medium (Effector:Target ratio (E:T) 10:1). Cancer cells only and cancer cells with the addition of Cisplatin (100 μM) (2251, Biotechne) were used as controls. Co-cultures were then incubated for 24 h at 37 °C in a normoxic incubator. Collected cells were then blocked with FcX and stained with anti-CD3-PE. DAPI was added immediately before acquisition to each sample to measure cancer cell death by DAPI uptake on a BD Fortessa using DIVA acquisition software.

## Adoptive cell transfer of T cells into tumor-naive and tumor-bearing mice

In total, 24 $Tcrd^{-/-}$ female mice received $2 \times 10^5$ E0771 cancer cells via orthotopic mammary fat pad transplantation as described above. Mice were split into 4 experimental groups with six age-matched mice each to receive either expanded Ly6C$^-$ γδT cells, Ly6C$^+$ γδT cells or naive, splenic CD8$^+$ T cells or PBS vehicle. 1 mouse did not develop a tumor. Adoptive T-cell transfer (ACT) was administered intravenously on 10, 13, 17, and 20 days post implantation of cancer cells. For each of the four rounds of ACT, cells were isolated from pooled LNs and spleens of 15 C57BL/6 J female mice; 14 samples were used for sorting Ly6C$^-$ and Ly6C$^+$ γδT cells and one set of LNs and spleen was used for CD8$^+$ T-cell isolation. Ly6C$^-$ and Ly6C$^+$ γδT cells were sorted and expanded as described above with supplementation of glucose (25 mM) (G7021, Sigma-Aldrich). CD8$^+$ T cells were isolated using MojoSort Mouse CD8 T Cell Isolation Kit (480035, BioLegend). CD8$^+$ T cells were cultured and expanded in a similar fashion to γδ T-cell populations. At 10 days post injection of cancer cells, experimental mice received 61,500 cells iv/mouse/group, at day 13 mice received 128,000 cells iv/mouse/group, at day 17 mice received 287,000 cells iv/mouse/group and at day 20 mice received 583,000 cells iv/mouse/group. Mice in control group received 100 μL PBS for each round of ACT. Once tumors reached 15 mm in any direction, mouse tissues were analyzed by flow cytometry.

For the in vivo conversion experiment, Ly6C$^-$ and Ly6C$^+$ γδ T cells were sorted from pooled LNs and spleens from three age-matched male WT C57BL/6 mice and three age-matched $Tcrb^{-/-}$ males or from six pooled $IL27ra^{-/-}$ mice. In total, $10^5$ cells (without expansion) in 100 μL PBS were injected intravenously into 3 or 5 NRG-SGM3 mice. Seven days after injections, tissues were processed for flow cytometry analysis.

## Telomere quantification

CD27$^+$Ly6C$^-$ and CD27$^+$Ly6C$^+$ γδ T cells from 6 pooled C57BL/6 mice per biological replicate were FACS-sorted as described above and were subsequently pelleted. Genomic DNA was extracted using QIAamp DNA Blood Mini kit (Qiagen, 51104) following the manufacturer's instructions. In all, 1 ng of DNA was used to proceed with the Absolute Mouse Telomere Length qPCR Assay Kit (ScienCell, M8919) according to the manufacturer's instructions with technical triplicates per sample.

## Western blot

Expanded γδ T-cell populations were stimulated with 50 ng/mL IL-27 (577402, BioLegend) for 30 min at 37 °C. Cell pellets were lysed in RIPA buffer (89901, ThermoFisher), supplemented with 1× HALT protease inhibitor (10085973, ThermoFisher Scientific) and 1x EDTA (1861274, ThermoFisher Scientific). Samples were loaded on pre-cast Bolt 4–12% Bis-Tris Plus (NW04122BOX, Thermo-Fisher Scientific) gradient gels and proteins were separated using 1× MOPS SDS Running Buffer (B0002, B0001, ThermoFisher Scientific) supplemented with Bolt Antioxidant (BT0005, Thermo-Fisher Scientific). Gels were transferred to iBlot2 NC Mini Stacks (IB23002, ThermoFisher Scientific) according to the manufacturer's instructions, and protein transfer onto nitrocellulose membrane was performed by using an iBlot2 (ThermoFisher Scientific) machine and pre-installed transfer program. After protein transfer, membranes were blocked with 5% milk in TBST on a shaker for 1 h. The following antibodies were purchased from Cell Signaling and used to probe blots: phospho-STAT1 (Tyr701, clone 58D6, 1:000), STAT1 (clone D1K9Y, 1:1000), phospho-STAT3 (Tyr705, D3A7, 1:2000), STAT3 (clone 124H6, 1:1000), phospho-STAT4 (Tyr693, clone D2E4, 1:1000), STAT4 (clone C46B10, 1:1000), HRP anti-mouse (1:2000), HRP anti-rabbit (1:2000). Anti-β-actin was purchased from Sigma (clone AC-74, 1:5000). Proteins were detected using Pierce ECL Western Blotting Substrate (32209, ThermoFisher Scientific) according to the manufacturer's instructions. Blots were exposed using a ChemiDoc imager (Bio-Rad). Quantification of western blot images was performed using Fiji Software by measuring optical density (OD) of inverted images. ODs of individual bands were corrected for background signal.

## Human γδ T cells

The West of Scotland Research Ethics Committee granted approval for use of human PBMCs for this study (ref. 20_WS_0066). Human PBMCs were obtained from whole blood samples from four healthy volunteers, drawn by a phlebotomist and collected in EDTA-coated tubes, after written consent. Up to 40 mL of undiluted whole blood was split into two tubes containing 15 mL of Pancoll (P04-60500, PAN-Biotech). Density gradient centrifugation to separate whole blood components was performed at 400 rcf for 30 min at room temperature with no break settings. The resulting top layer (serum) was carefully taken off to facilitate the collection of PBMCs. PBMCs were transferred into a new tube and washed with PBS. Centrifugation at 400 rcf for 10 min at room temperature with breaks was performed twice in order to remove excess Pancoll. Cells were then counted by Trypan Blue exclusion with a hemocytometer. In all, 24-well plates have been coated with 1μg purified human TCRγ/δ antibody (331202, BioLegend) per well overnight at 4 °C. $3 \times 10^5$ PBMCs per well were added in RPMI-1640 medium (R8758, Sigma-Aldrich) supplemented with 10% FCS, 100 U/mL penicillin, 100 μg/mL streptomycin, 2 mM gluta-mine, 1× non-essential amino acids (NEAA) (11140-035, Gibco), 1× essential amino acids (EAA) (11130-036, Gibco), 50 μM

2-mercaptoethanol (βME), (31350-101, Gibco), 10 mM HEPES, 1 mM sodium pyruvate (S8636, Sigma), 1000 IU/mL recombinant (r) human (h) IL-2 (200-02, PeproTech), and 10 ng/mL rhIL-15 (200-15, PeproTech). PBMCs were split every 3 days for a duration of two weeks and seeded at a density of $3 \times 10^5$ cells per well (24-well plate) or at later stages as of $2 \times 10^6$ cells per well (six-well plate). Where indicated, PBMCs received 50 ng/mL rhIL-27 (200-38, PeproTverech) throughout expansion. These experiments were independently verified at Sanquin (Amsterdam, Netherlands) with a similar protocol. Isolated PBMCs were seeded in a 24-well plate at $3 \times 10^5$ PBMCs per well with $5 \times 10^6$ irradiated PBMCs pooled from 15 healthy blood donors (feeder cells) and 1 µg/ml purified α-TCR γ/δ (331202, BioLegend), in RPMI supplemented with 5% FCS, 5% HS, 1000 IU/ml IL-2 and 10 ng/ml IL-15 (200-15, PeproTech), with six wells in total for each patient. Three wells per patient received an additional 50 ng/ml IL-27 (200-38, PeproTech). Medium was refreshed typically every other day and when a monolayer at the bottom of the well was observed, cells were passaged. After 14 days, cells were harvested, washed, and manually counted with trypan blue.

## Statistical analysis and data visualization

Statistical significance was calculated using GraphPad Prism version 9.0.2. As indicated in individual figure legends, comparison between two groups was determined by either paired or unpaired Student's *t* test. Multiple experimental groups were analyzed with one-way ANOVA followed by Tukey's post hoc test. Tumor-related survival was plotted with Kaplan–Meier survival curves and analyzed using Mantel–Cox (Log-rank) test. If not otherwise mentioned, data are represented as mean ± standard deviation (SD). Sample size (*n*) for each experiment is indicated in figure legends.

# Data availability

The datasets used and produced in this study are available in the following databases: scRNAseq data: Gene expression. ArrayExpress #E-MTAB-10677. RNA-Seq data: Gene expression. ArrayExpress #E-MTAB-13897.

The source data of this paper are collected in the following database record: biostudies:S-SCDT-10_1038-S44318-024-00133-1.

# Peer review information

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

## Acknowledgements

The authors thank Jos Jonkers (Netherlands Cancer Institute) for mammary tumor mouse models, Kristina Kirschner (University of Glasgow) for assistance with scRNAseq data, and Catherine Winchester (CRUK Scotland Institute) for advice. We thank the Core Services and Advanced Technologies facilities at the Cancer Research UK Scotland Institute. RW was supported by Breast Cancer Now (2019DecPhD1349 awarded to SBC). SCE and AK were supported by Cancer Research UK Glasgow Cancer Centre (C596/A25142). Tenovus Scotland (S17-17 awarded to SBC) provided funding for the study. SBC was supported by the Annie McNab Bequest (CRUK Scotland Institute). KB, CM, HH, CH, AH, AM, RS, DW, and PJW were supported by Cancer Research UK (A31287, A29799, A29801, DRCPFA-Nov21\100001). CH and AM were supported by CCLG Little Princess Trust Project Grant CCLGA (2020 24). GJG and AH were supported by Wellcome Trust (217093/Z/19/Z) and the Medical Research Council (MR/V010972/1). GWJ and DGH were supported by Versus Arthritis (22706). DJP and NS were supported by the Biotechnology and Biological Sciences Research Council (BB/R017808/1). MCW, SMC, and LW were supported by Sanquin.

## Author contributions

**Robert Wiesheu:** Conceptualization; Data curation; Formal analysis; Supervision; Validation; Investigation; Visualization; Methodology; Writing—original draft; Project administration; Writing—review and editing. **Sarah C Edwards:** Conceptualization; Formal analysis; Investigation; Methodology; Writing—review and editing. **Ann Hedley:** Formal analysis; Visualization; Writing—review and editing. **Holly Hall:** Formal analysis; Visualization. **Marie Tosolini:** Formal analysis; Investigation; Visualization; Writing—review and editing. **Marcelo Gregorio Filho Fares da Silva:** Formal analysis; Investigation. **Nital Sumaria:** Formal analysis; Investigation; Visualization; Writing—review and editing. **Suzanne M Castenmiller:** Formal analysis; Validation; Investigation; Writing—review and editing. **Leyma Wardak:** Formal analysis; Investigation. **Yasmin Optaczy:** Formal analysis; Investigation. **Amy Lynn:** Investigation. **David G Hill:** Resources. **Alan J Hayes:** Resources. **Jodie Hay:** Resources. **Anna Kilbey:** Formal analysis; Validation; Investigation; Methodology; Writing—review and editing. **Robin Shaw:** Formal analysis. **Declan Whyte:** Investigation. **Peter J Walsh:** Formal analysis. **Alison M Michie:** Resources; Writing—review and editing. **Gerard J Graham:** Resources. **Anand Manoharan:** Resources. **Christina Halsey:** Resources. **Karen Blyth:** Resources; Project administration. **Monika C Wolkers:** Supervision; Validation; Writing—review and editing. **Crispin Miller:** Formal analysis; Supervision. **Daniel J Pennington:** Resources; Supervision; Writing—review and editing. **Gareth W Jones:** Resources. **Jean-Jacques Fournie:** Data curation; Formal analysis; Supervision; Visualization. **Vasileios Bekiaris:** Resources; Formal analysis; Supervision; Investigation; Writing—review and editing. **Seth B Coffelt:** Conceptualization; Resources; Data curation; Formal analysis; Supervision; Funding acquisition; Validation; Investigation; Visualization; Methodology; Writing—original draft; Project administration; Writing—review and editing.

Source data underlying figure panels in this paper may have individual authorship assigned. Where available, figure panel/source data authorship is listed in the following database record: biostudies:S-SCDT-10_1038-S44318-024-00133-1.

## Disclosure and competing interests statement

The authors declare no competing interests.

# Expanded View Figures

**Figure EV1. Expression of cytotoxic markers and TCR usage by CD27$^+$Ly6C$^-$ and CD27$^+$Ly6C$^+$ γδ T cells.**

(**A**) Flow cytometry plots for expression of indicated proteins in CD27$^+$Ly6C$^-$ and CD27$^+$Ly6C$^+$ γδ T cells from spleen of naive FVB/n mice. Fluorescence minus one (FMO) controls were used to set gating. (**B**) Flow cytometry plots of TCR chain usage on CD27$^+$ γδ T cells and Ly6C expression of each population.

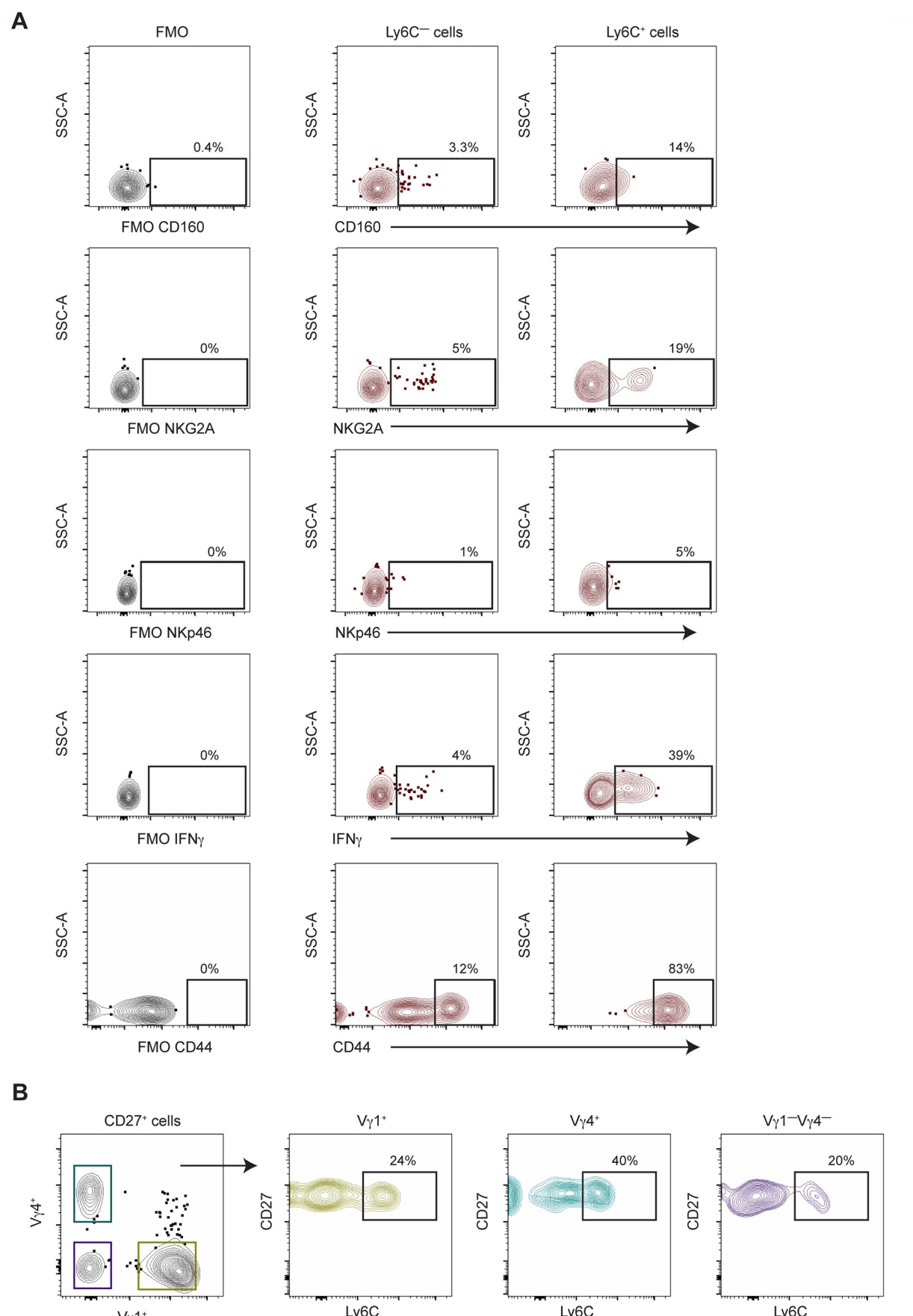

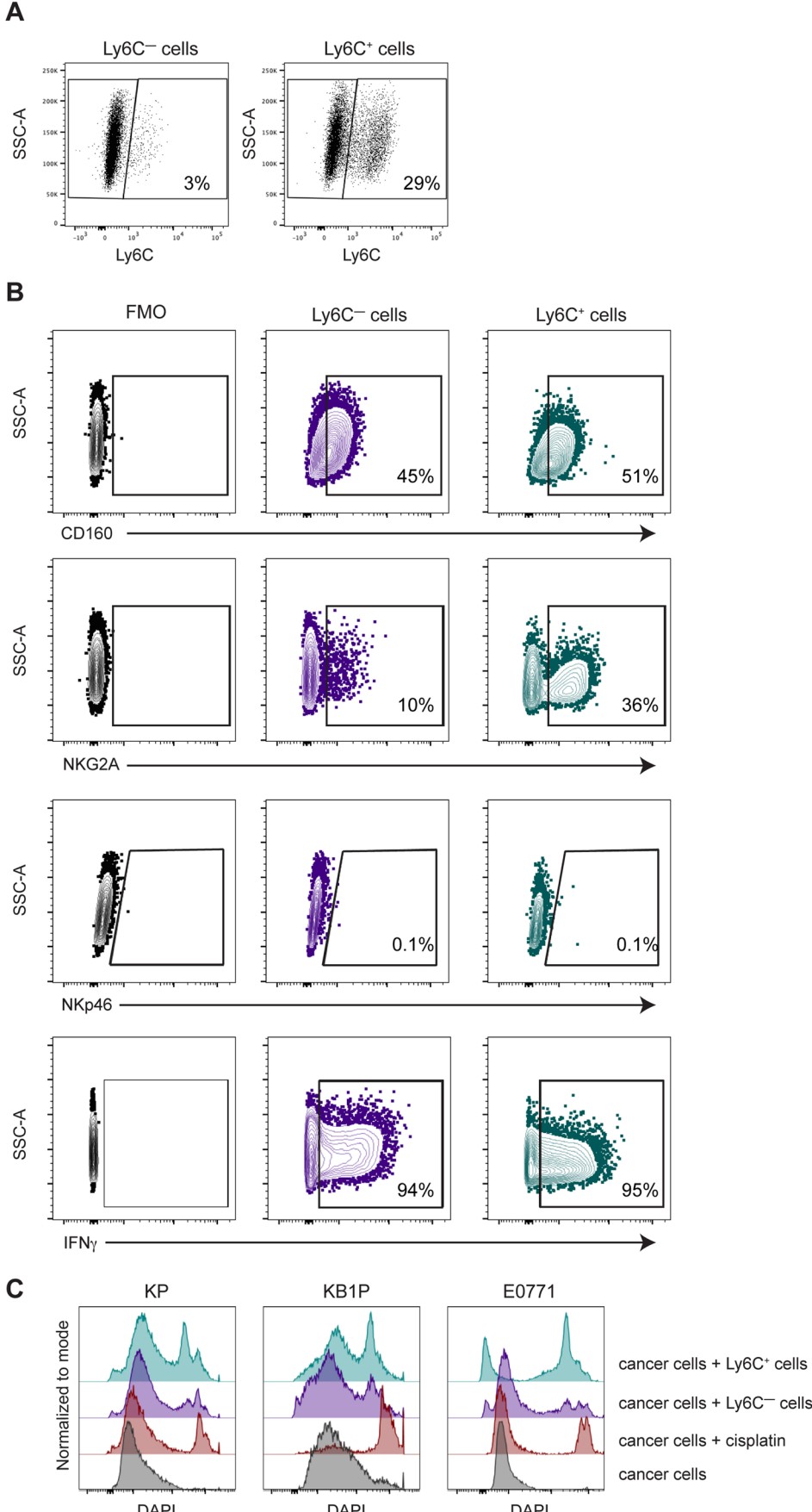

◀  **Figure EV2.  Phenotyping and cancer cell killing ability of CD27$^+$Ly6C$^-$ and CD27$^+$Ly6C$^+$ γδ T cells.**

(**A**) Flow cytometry plots of Ly6C expression on sorted CD27$^+$Ly6C$^-$ and CD27$^+$Ly6C$^+$ γδ T-cell subsets expanded ex vivo over 4 days in the presence of CD3/CD28 Dynabeads, IL-2, and IL-15. (**B**) Flow cytometry plots for expression of indicated proteins in expanded CD27$^+$Ly6C$^-$ and CD27$^+$Ly6C$^+$ γδ T cells. Fluorescence minus one (FMO) controls were used to set gating. (**C**) Representative histograms of cancer cell death measured by DAPI uptake after co-culture with ex vivo-expanded CD27$^+$Ly6C$^-$ and CD27$^+$Ly6C$^+$ γδ T cells or cisplatin treatment.

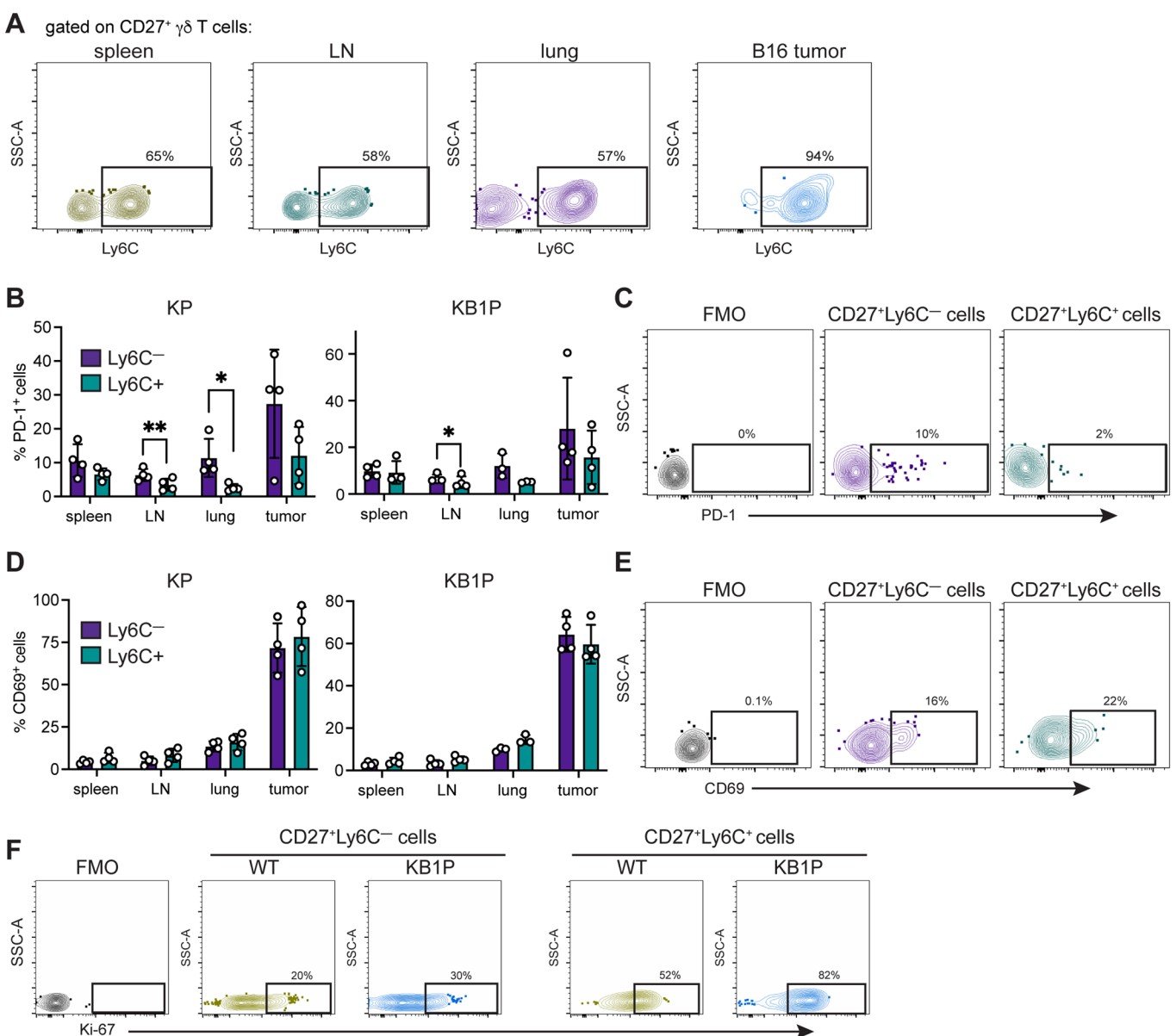

**Figure EV3. Ly6C and Ki-67 expression in tumor-associated CD27+Ly6C− and CD27+Ly6C+ γδ T cells.**

(A) Flow cytometry plots of Ly6C expression on CD27+ γδ T cells in indicated tissue from B16-F1 tumor-bearing mice. (B) Frequency of PD-1+ cells in CD27+Ly6C− and CD27+Ly6C+ γδ T cells in indicated tissues of tumor-bearing KP and KB1P mice (n = 3–4). Each dot represents one tumor-bearing mouse. *P < 0.05, **P < 0.01 (paired t test). Data are represented as mean ± SD. (C) Representative flow cytometry plots of PD-1 expression on CD27+Ly6C− and CD27+Ly6C+ γδ T cells from lungs of tumor-bearing KP mice. (D) Frequency of CD69+ cells in CD27+Ly6C− and CD27+Ly6C+ γδ T cells in indicated tissues of tumor-bearing KP and KB1P mice (n = 3–4). Each dot represents one tumor-bearing mouse. Data are represented as mean ± SD. (E) Representative flow cytometry plots of CD69 expression on CD27+Ly6C− and CD27+Ly6C+ γδ T cells from lungs of tumor-bearing KP mice. (F) Flow cytometry plots of Ki-67 expression on CD27+Ly6C− and CD27+Ly6C+ γδ T cells from FVB/n WT and KB1P tumor-bearing mice.

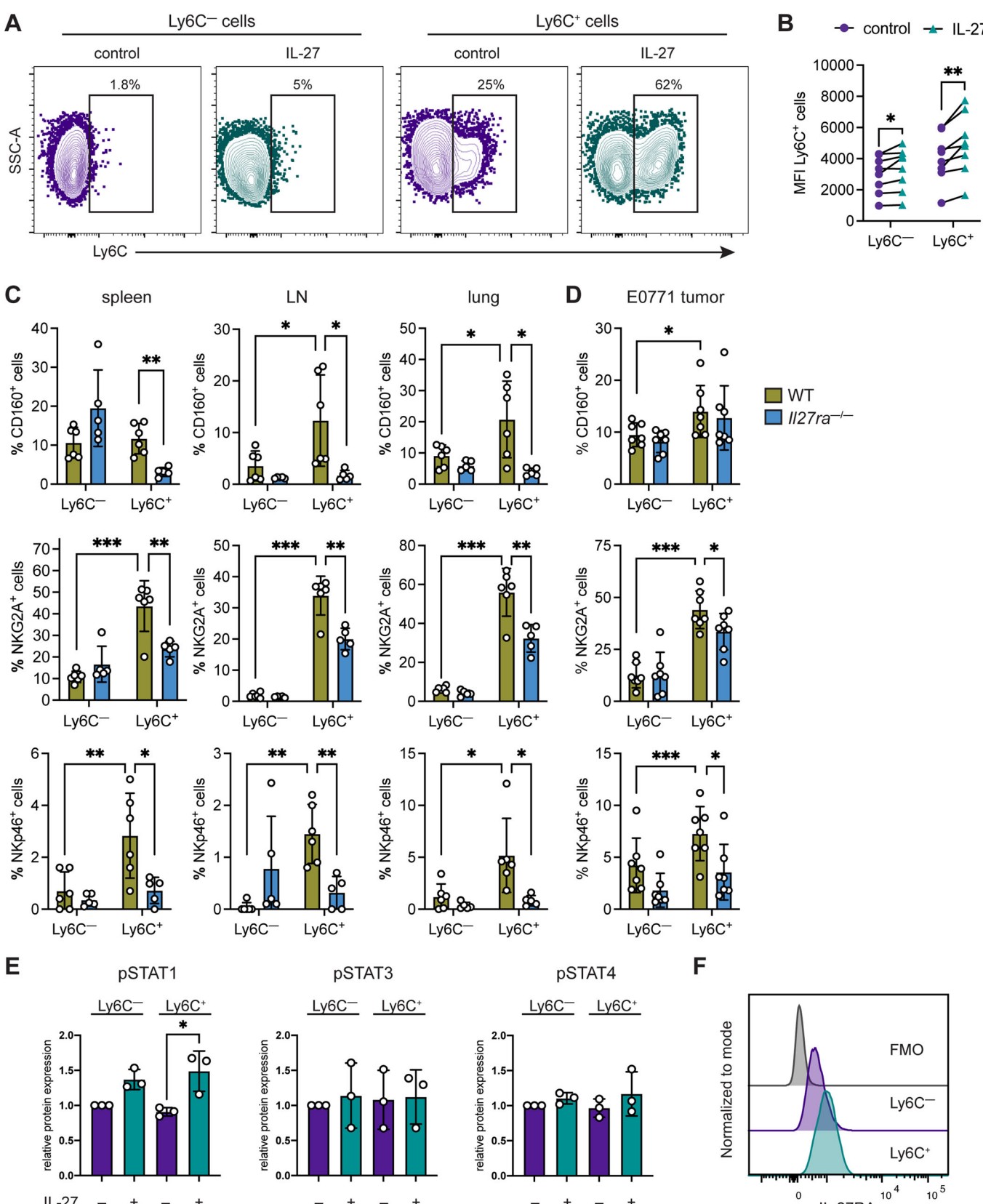

◀  **Figure EV4.  IL-27 regulates CD27⁺Ly6C⁺ γδ T cells.**

(A) Flow cytometry plots of Ly6C expression on sorted CD27⁺Ly6C⁻ and CD27⁺Ly6C⁺ γδ T-cell subsets expanded ex vivo over 4 days in the presence of CD3/CD28 Dynabeads, IL-2, and IL-15, as well as IL-27 where indicated. (B) Median fluorescence intensity (MFI) of Ly6C expression after gating on Ly6C⁺ cells within sorted CD27⁺Ly6C⁻ or CD27⁺Ly6C⁺ γδ T-cell subsets expanded ex vivo over 4 days with CD3/CD28 beads, IL-2, and IL-15 (control), with IL-27 as indicated. Individual replicates are shown as pairs ($n = 8$). Each dot represents expanded cells from pooled LNs and spleens of 6 mice. *$P < 0.05$, **$P < 0.01$ (paired $t$ test). (C, D) Proportion of CD160, NKG2A, and NKp46-expressing Ly6C⁻ and Ly6C⁺ cells in indicated tissues from C57BL/6 WT ($n = 6$ tumor-free, 7 tumor-bearing) or *Il27ra⁻/⁻* ($n = 5$ tumor-free, 7 tumor-bearing) mice. Each dot represents one mouse *$P < 0.05$, **$P < 0.01$, ***$P < 0.001$ (unpaired $t$ test). Each dot represents one mouse. Data are represented as mean ± SD. *$P < 0.05$ (unpaired or paired student $t$ test). (E) Densitometry graphs representing relative protein expression of indicated phosphorylated (p) STAT proteins after in vitro culture of CD27⁺Ly6C⁻ and CD27⁺Ly6C⁺ γδ T cells in the presence or absence of IL-27 from cells from (A). First condition was set to 1 in order to normalize between independent biological replicates ($n = 3$). Each dot represents one independent in vitro culture from a pool of 6 mice. Data are represented as mean ± SD. *$P < 0.05$ (repeated measures one-way ANOVA followed by Tukey's posthoc test). (F) Representative histograms of IL-27RA expression in CD27⁺Ly6C⁻ and CD27⁺Ly6C⁺ γδ T cells from lymph node tissue of C57BL/6 WT mice.

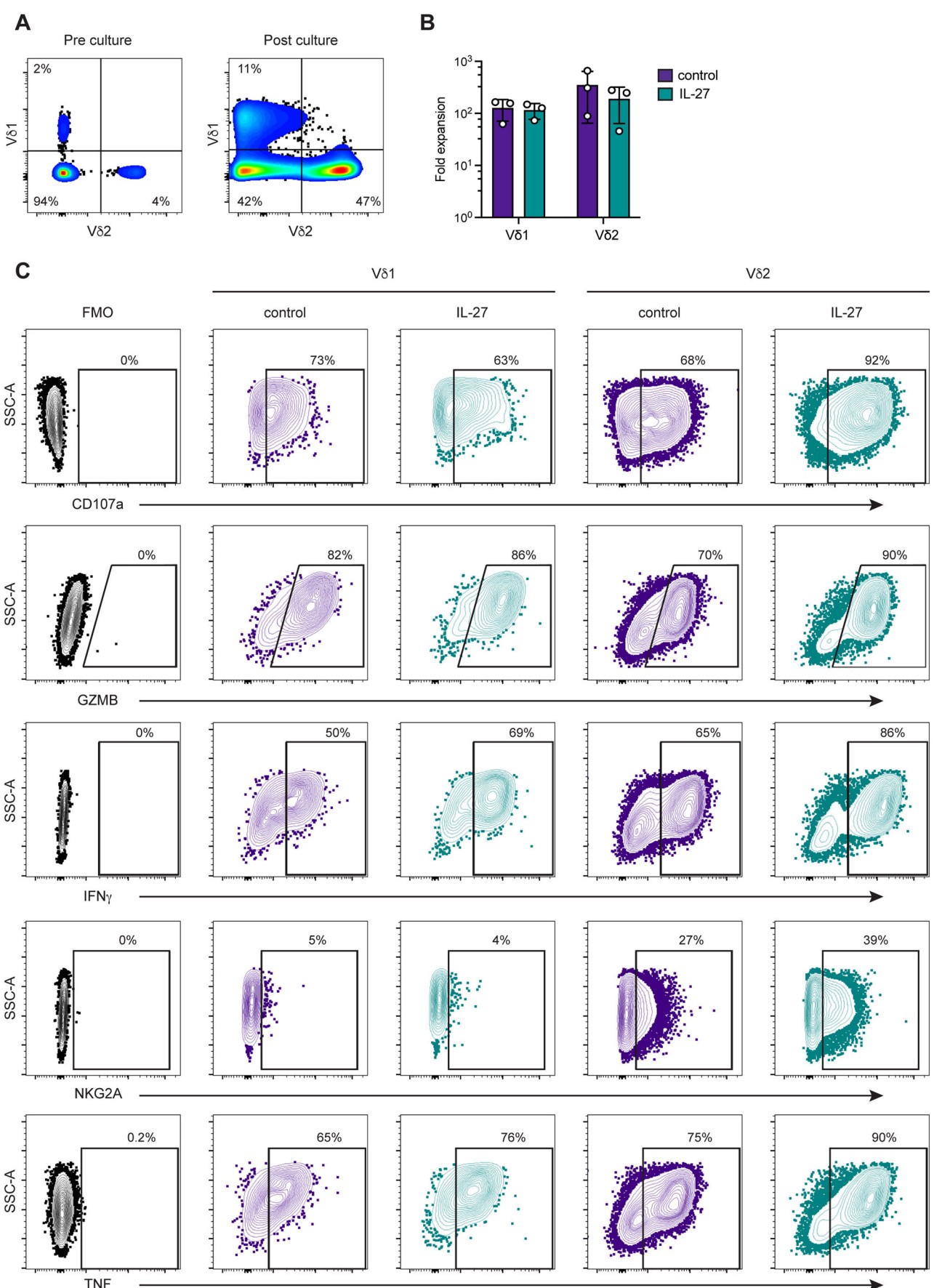

◄ **Figure EV5. Human Vδ2 cells respond to IL-27 stimulation.**

(A) Flow cytometry plots of Vδ1 and Vδ2 T cells before expansion (left) and after culture for 14 days with IL-2 and IL-15. (B) Fold expansion of human Vδ1 and Vδ2 cells with IL-2 and IL-15 (control) or IL-2, IL-15, and IL-27 ($n = 3$ human PBMC donors/group). Data are represented as mean ± SD. (C) Flow cytometry plots of ex vivo-expanded cells from (A). Live CD3$^+$ cells were gated on Vδ1 and Vδ2. Expression of CD107a, Granzyme B (GZMB), IFNγ, NKG2A and TNF was measured on Vδ1 and Vδ2 cells for both culture conditions. FMO controls were used to set gating.

