## [Peer Review File · The EMBO Journal]

IL-27 maintains cytotoxic Ly6C⁺ gamma delta T cells that arise from immature precursors

Robert Wiesheu, Sarah Edwards, Ann Hedley, Holly Hall, Marie Tosolini, Marcelo Fares da Silva, Nital Sumaria, Suzanne Castenmiller, Leyma Wardak, Yasmin Optaczy, Amy Lynn, David Hill, Alan Hayes, Jodie Hay, Anna Kilbey, Robin Shaw, Declan Whyte, Peter Walsh, Alison Michie, Gerry Graham, Anand Manoharan, Christina Halsey, Karen Blyth, Monika Wolkers, Crispin Miller, Dan Pennington, Gareth Jones, Jean-Jacques Fournie, Vasileios Bekiaris, and Seth Coffelt

Corresponding author: Seth Coffelt (seth.coffelt@glasgow.ac.uk)

Review Timeline:

Submission Date:	12th Mar 24
Editorial Decision:	13th Apr 24
Revision Received:	19th Apr 24
Accepted:	3rd May 24

Editor: Kelly Anderson

Transaction Report:

Dear Reviewers,

We appreciate your thorough analysis and helpful suggestions on our study. These comments have been instrumental in strengthening our conclusions. Since you last read the paper, quite a lot has changed. We have added several new and exciting pieces of data about the biology of CD27+ $\gamma\delta$ T cells. These additions are summarized here:

1. Use of T-Bet reporter mice to confirm that CD27⁺Ly6C⁺ $\gamma\delta$ T cells express T-Bet, the IFN γ -inducing transcription factor, while CD27⁺Ly6C⁻ cells do not (Figure 2D)
2. Gene expression analysis of V γ 1⁺ and V γ 4⁺ cells based on their Ly6C protein expression (Figure 3)
3. Phenotypic and functional characterization of CD27⁺Ly6C⁻ and CD27⁺Ly6C⁺ $\gamma\delta$ T cell subsets during *in vitro* expansion (Figure 4A-F)
4. Differential cancer killing ability of CD27⁺Ly6C⁻ and CD27⁺Ly6C⁺ $\gamma\delta$ T cell subsets both *in vitro* and *in vivo* (Figure 4G-K)
5. Inclusion of other cancer models showing the frequency of CD27⁺Ly6C⁺ $\gamma\delta$ T cells in various organs and tumors as well as phenotype comparison of the subsets (Figure 5)
6. Ability of CD27⁺Ly6C⁻ $\gamma\delta$ T cells to convert into CD27⁺Ly6C⁺ $\gamma\delta$ T cells and the phenotype of recovered cells in lymphopenic mice (Figure 6)
7. Telomere length of CD27⁺Ly6C⁻ and CD27⁺Ly6C⁺ $\gamma\delta$ T cell subsets (Figure 6I)
8. Identification of IL-27 as a cytokine that maintains Ly6C expression and cytotoxic profile with mechanistic insight into signalling cascades (Figure 7)
9. Cross-species comparison and validation of the effects of IL-27 on human $\gamma\delta$ T cell subsets (Figure 8).

Below we provide a point-by-point rebuttal to your questions and comments in blue. Thank you for taking the time to improve our work.

Sincerely,
Seth Coffelt (on behalf of all the co-authors)

Referee #1:

EMBOJ -2020-106160

Ly6C defines a subset of memory-like CD27+ $\gamma\delta$ T cells with inducible cancer-killing function

CD27neg IL-17-producing $\gamma\delta$ T cells promote cancer progression and metastasis while CD27+ IFN γ -producing $\gamma\delta$ T cells have well-established anti-tumour effects. The intricacies of their anti-cancer functions are however less well understood.

The interesting study by Wiesheu et al., aimed to elucidate phenotype and heterogeneity of CD27+ $\gamma\delta$ T cells in mice. The authors performed single-cell RNA sequencing (scRNA-Seq) on $\gamma\delta$ T cells from the lung of healthy mice. They focused computationally on the analysis of 468 CD27+ $\gamma\delta$ T cells and identified 3 different clusters. While cluster 2 likely represents contaminating IL-17-producing $\gamma\delta$ T cells, cluster 0 represents a subset with increased cytotoxic ability and cluster 1 a more naïve subset.

Importantly, the authors identified Ly6C as a potential marker discriminating between naïve and memory-like subsets within the CD27⁺ IFN- γ -producing $\gamma\delta$ T cell subset in the periphery tissues including lung, lymph nodes and spleen. Although in vivo depletion of V γ 1 T cells, which account for a subset of CD27⁺ Ly6C⁺ $\gamma\delta$ T cells, had no effect on the growth of breast cancer in the KB1P model, bulk $\gamma\delta$ T cells could be expanded ex vivo and were cytotoxic to an array of cancer cell lines in vitro.

Overall, the study is well written and results shown are interesting; especially the scRNA-Seq analysis provides the field of $\gamma\delta$ T cell biology with (i) an useful resource of scRNA-Seq data to characterize the compositional diversity of CD27⁺ $\gamma\delta$ T cells in the mouse, and (ii) an insight linking mouse $\gamma\delta$ T cell subsets to human $\gamma\delta$ T cell subsets.

However, further experiments and controls are needed to support the conclusions on Ly6C made by the authors.

Major points:

1. Fig. 1: The authors claim that Cluster 2 is most likely a lung-resident $\gamma\delta$ T cell subset that expresses high mRNA levels of CD27 but lacks CD27 protein expression. This seems unusual. How is the transcriptome of cluster 2 compared to CD27^{neg} IL-17-producing $\gamma\delta$ T cells? Is there a precedent in the literature? Could a more stringent doublet exclusion help?

Response: We appreciate the Reviewer's suggestions around this observation. The transcriptome of Cluster 2 is identical to V γ 6 cells (see Edwards *et al.*, *J Exp Med* 2023). The bioinformatic pipeline used to analyze scRNAseq data here excludes two or more cells captured in a single droplet so doublets are excluded. Other labs have reported *Cd27* transcripts in IL-17-producing cells corroborating our results (Tan *et al.*, *Cell Reports* 2019).

2. Do CD27^{neg} IL-17-producing $\gamma\delta$ T cells (more memory-like phenotype) express Ly6C at higher level compared with CD27⁺ $\gamma\delta$ T cells? Is Ly6C also marking a subset of memory-like cells in IL-17-producing $\gamma\delta$ T cell subset?

Response: We provide the data to this question below. CD27⁺ $\gamma\delta$ T cells express higher levels of Ly6C than CD27⁻ $\gamma\delta$ T cells in lymph nodes of female FVB/n mice. As the focus of the study is on CD27⁺ $\gamma\delta$ T cells, we have not included this comparison in the manuscript.

3. Fig. 4A: In order to make a statement about the frequency of $\gamma\delta$ T cell subsets in control and

KB1P mice, the authors have to show actual cell numbers of these cells normalized to mg of tissue assessed. CD27+ $\gamma\delta$ T cells in KB1P mice have higher percentages of Ki67 (Fig. 4D) and it is likely that they are increased in numbers. Please also specify what lymph nodes (LNs) were examined in control and tumour-bearing mice and indicate whether these were draining or non-draining in the tumour-bearing KB1P mice.

Response: In revised Figure 5C, we show that Ki-67 expression changes between tumor-free control mice and tumor-bearing KB1P mice. However, absolute numbers do not change. It is not clear why the suggested proliferation by Ki-67 staining does not correlate with increased numbers. Perhaps the cells leave the lymph node or Ki-67 is not a good marker for proliferation in these cells. We have modified our description of the data in the text. Axillary, brachial, and inguinal lymph nodes were pooled for analysis. These included both draining and non-draining lymph nodes.

4. Fig. 4: In order to establish the effect of $\gamma\delta$ T cell subsets on the development of mammary tumours, the authors have to show $\gamma\delta$ T cell infiltration in tumours of KB1P mice. Pooling of tumours or concatenating FACS results from different tumours computationally will help with the scarcity issue the authors encountered. If $\gamma\delta$ T cell infiltration is negligible in the tumours of KB1P mice, why do the authors use this model? Is $\gamma\delta$ T cell infiltration in tumours higher in the mice with engrafted syngeneic KB1P tumours (used in Fig. 5) and should they be used for the analysis in Fig. 4?

Response: These data are now presented in Fig. 5A. CD27+ $\gamma\delta$ T cells are readily found in KB1P tumors and other cancer models.

5. Fig. 4: In other tumour models, it has been shown that activation and expansion of $\gamma\delta$ T cells in the tumour-draining LNs is crucial for the infiltration of $\gamma\delta$ T cells in the tumour microenvironment. The authors should stain CD27+ Ly6C+ and CD27+ Ly6C- $\gamma\delta$ T cells in the tumour-draining LNs and the tumour for activation markers like CD25, CD69 or PD-1 to confirm whether KB1P tumours can trigger a $\gamma\delta$ T cell response in the peripheral lymphoid tissues.

Response: We have performed these experiments as requested. PD1 and CD69 data are presented in Extended Data Fig. 3B-E. PD1 is not expressed on CD27+ $\gamma\delta$ T cells at high levels, but Ly6C⁻ cells tend to express slightly higher levels. CD69 is not present to a high degree until CD27+ $\gamma\delta$ T cells enter tumors – both Ly6C+/- subsets express equivalent amounts. We have not included data for CD25 because the cells do not really produce this molecule. We provide an example below showing that the antibody works well on conventional T cells from lymph nodes, but does not stain CD27+ $\gamma\delta$ T cells.

6. Fig. 5: It is surprising to the reviewer that the authors have chosen to deplete V γ 1 T cells in vivo. This approach is neither specific to their newly identified "cytotoxic" CD27⁺ Ly6C⁺ subset nor targets all CD27⁺ cells. In addition, the depletion of V γ 1⁺ T cells achieved in the lung is not sufficient to support the conclusion made by the authors. Were V γ 1⁺ T cells more efficiently depleted in other tissues, especially peripheral lymphoid tissues and tumour? In order to draw firm conclusions, the authors should perform a more informative in vivo experiment, namely the adoptive transfer of either CD27⁺ Ly6C⁺ or CD27⁺ Ly6C⁻ control $\gamma\delta$ T cells (sorted after in vitro expansion if enough cells can't be obtained from healthy animals) into KB1P tumor-bearing mice. Homing and activation of these two subsets of $\gamma\delta$ T cells should be examined in both peripheral lymphoid tissues and in the tumour.

Response: We wholeheartedly agree with the Reviewer on this point. The V γ 1 data have been removed from the manuscript because Ly6C⁺/⁻ cells are made up of both V γ 1 and V γ 4 cells (Figure 3B). We performed the suggested experiment, and it is presented in Fig. 4H-K. The data show that Ly6C⁺ cells are able to control E0771 tumor growth.

7. Fig. 6: Here the authors expanded bulk $\gamma\delta$ T cells in vitro and by doing so missed the chance to carefully characterize the inherent cytotoxic phenotype of CD27⁺ Ly6C⁺ and CD27⁺ Ly6C⁻ $\gamma\delta$ T cells without co-culture 'artefacts'. Cleaner experiments should be performed expanding sorted CD27⁺ Ly6C⁺ and CD27⁺ Ly6C⁻ $\gamma\delta$ T cells (along with specific V γ chains if possible) separately. MFIs of IFN- γ and granzyme B should be shown.

Response: As suggested, we performed these experiments and they are presented in Fig. 4. Killing assays are shown in Fig. 4G. We were not able to accomplish killing assays with V γ 1 and V γ 4 cells.

8. Fig. 7: Killing assays should be performed separately with both expanded CD27+ Ly6C+ and CD27+ Ly6C- $\gamma\delta$ T cells to definitively answer the question whether CD27+ Ly6C+ $\gamma\delta$ T cells - with a memory-like phenotype and higher expression of killer-associated molecules - do show faster/stronger cytotoxicity against cancer cells compared to their CD27+ Ly6C- counterparts. Beyond the scope of this paper but of great interest would be to determine the mode of killing by CD27+ Ly6C+ and CD27+ Ly6C- $\gamma\delta$ T cells using blocking antibodies (IFN- γ , $\gamma\delta$ TCR, FasL etc).

Response: This was an important suggestion. We performed this experiment and it is presented in Fig. 4G. For the Reviewer's curiosity, we have preliminary data suggesting that CD27+Ly6C+ cells use FASL but not NKG2D for killing cancer cells (data are not included in this study). This needs further interrogation.

Minor points:

1. Please add the actual percentage of each gate shown in the representative FACS plots.

Response: This has been provided in Extended Data Figures 1-5.

2. Figure 2A and B: Instead of showing the expression of Ly6C and CCR7 by CD27+ $\gamma\delta$ T cells separately, please show the expression of CCR7 by CD27+ Ly6C+ and CD27+ Ly6C- $\gamma\delta$ T cells.

Response: As shown in Fig. 2, CCR7 is expressed by Ly6C- cells but not Ly6C+ cells. We have confirmed that the antibody works well on dendritic cells by comparing migratory DCs in lymph nodes (red histogram below) versus lymph node resident DCs (blue histogram) versus FMO. Therefore, we believe the CCR7 staining on $\gamma\delta$ T cells is real and that these cells just fail to express high levels. For this reason, we did not continue to use CCR7 as a marker of immature CD27+Ly6C- $\gamma\delta$ T cells.

3. Figure 2C: Please show representative FACS plots showing the expression of CD160, NKG2A, NKp46 and IFN- γ by CD27+ Ly6C- $\gamma\delta$ T cells in the same panel for direct comparison between CD27+ Ly6C+ and CD27+ Ly6C- $\gamma\delta$ T cells.

Response: This has been provided in Extended Data Figure 1.

4. Fig. 3A: Please show the representative FACS plot of CD27+ Ly6C- $\gamma\delta$ T cells in the same panel for direct comparison between CD27+ Ly6C+ and CD27+ Ly6C- $\gamma\delta$ T cells.

Response: This has been provided in Extended Data Figure 2.

5. Fig. 3E: please show representative FACS plots for Ly6C+ and Ly6C- $\gamma\delta$ T cells.

Response: This has been provided in Extended Data Figure 3.

6. Figure 4C: Labels for y-axis is wrong. It should be: % of CD27+ Ly6C- $\gamma\delta$ T cells and % of CD27+ Ly6C+ $\gamma\delta$ T cells for the left and right panels, respectively.

Response: This has been corrected.

7. Fig. 5: Please use K14-Cre+ Brca1WT, Trp53WT mice as another WT control to control for Cre toxicity.

Response: We understand the Reviewer's concerns about using proper controls. However, in this case, we feel that this request does not concur with the 3Rs. In order to generate these mice, we would need to cross KB1P with FVB/n mice for three generations, using multiple mice. Moreover, we have never observed any toxicity from Cre expression. Jos Jonkers' lab who generated the breast cancer models has also never observed toxicity from Cre expression.

8. Fig. 5: Do syngeneic recipients of KB1P tumour transplants develop lung metastasis as has been reported for the K14cre;Cdh1F/F;Trp53F/F model (Doornabel et al., 2012)? Has metastatic load been taken into account when the composition and activation of CD27+ Ly6C+ and CD27+ Ly6C- $\gamma\delta$ T cells was assessed in the lung?

Response: Yes the KB1P tumor transplants metastasize to the lung (see Edwards *et al. J Exp Med* 2023). Here, lungs were analyzed before the appearance of metastasis.

9. Material and Methods: FACS - description for live/dead stain is repeated. Delete one.

Response: Thank you for pointing this out – it has been corrected.

Referee #2:

In this manuscript, Wiesheu, Coffelt and colleagues have focused on CD27+ gd T cells in lung, spleen and lymph nodes. They identified two different subsets of CD27+ gd T cells using scRNA-seq which can be distinguished by the expression of Ly6C. In their breast cancer mouse model, CD27+ gd T cells do not seem to play any role. However, after their expansion in vitro, they become more cytotoxic and kill mammary cancer cell lines. They also show that

CD27+Ly6C+ gd T cells show more cytotoxic phenotype while CD27+Ly6C- gd T cells are more naïve. Importantly, the authors have also performed a cross-species analysis and have found that Ly6C+ mouse gd T cells resemble more mature human lymphocytes while Ly6C- cells resemble more naïve cells in humans.

Although the study reveals novel biology of CD27+ gd T cells, there are several major concerns that authors need to address to make this study suitable for publication in the respected EMBO Journal appealing to wide readership.

Major concerns

1. The authors computationally separated CD27+ gd T cells based on Cd27 mRNA expression. Later they conclude that cluster 2 has Cd27 mRNA but no CD27 protein expression revealing the discrepancy or sometimes non-correlated nature of mRNA and protein levels in a cell. Authors provide no direct evidence that computationally segregated (based on Cd27 mRNA) CD27+ gd T cells are indeed bonafide CD27+ gd T cells. The authors should separately sequence CD27+ gd T cells using CD27 antibody to validate that the cells they are characterizing are indeed bonafide CD27+ gd T cells.

Response: CD27+ and CD27- \$\gamma\delta\$ T cells are well established subsets in the field, having been first described by Bruno Silva-Santos and colleagues in 2009 (*Nature Immunology*). We provide further RNAseq data regarding the transcriptome of CD27+ \$\gamma\delta\$ T cells in revised Figure 3. The results confirm the scRNAseq data where Ly6C- cells are transcriptionally different to Ly6C+ cells.

2. I have a major concern regarding the use of Ly6C as a marker to distinguish cytotoxic/memory cells from the naïve cells. The claim that Ly6C is a good marker of CD27+ cytotoxic gd T cells is based on the staining of few marker genes such as CD160, NKG2A, NKp46 and IFN- γ . However, it is very clear that not all cells in the CD27+ Ly6C+ compartment express these markers, it is noteworthy that the proportion of cells expressing these markers is - most of the time - not even reaching up to 50% which means half of the cells are not even expressing these markers. So how can authors confidently say that Ly6C separates cluster 0 from cluster 1? In my view, there is a simple solution to this problem - the authors should perform scRNA-seq on CD27+ Ly6C+ and CD27+ Ly6C- gd T cells after FACS sorting and analyze/integrate this data with pan CD27+ gd T cells data (concern 1) to validate that the two clusters can indeed be separated using Ly6C. Figure panel 2 is not convincing enough to claim that Ly6C separates cytotoxic cells from naïve cells.

Response: To address the Reviewer's concerns, we provide multiple lines of evidence:

- 1. Multiple scRNAseq datasets have made similar findings with regards to Ly6C expression (du Halgouet et al, 2024; Li et al, 2022; McIntyre et al, 2020; Park et al., 2021; Tan et al, 2019; Yang et al, 2023)
- 2. We now provide bulk RNAseq analysis of CD27+Ly6C- and Ly6C+ \$\gamma\delta\$ T cells together with specific TCR-defined populations in Fig. 3
- 3. In Fig. 2, we highlight the phenotypic difference between CD27+Ly6C- and Ly6C+ \$\gamma\delta\$ T cells at the protein level. We provide additional data showing that Ly6C+ cells express higher levels of T-Bet.
- 4. Fig 4 shows functional differences between the subsets
- 5. Fig. 6 shows that Ly6C- cells convert to Ly6C+ cells and there is a difference in telomere length between the subsets indicating that Ly6C+ cells have undergone more cell cycle events.

3. The authors should characterize the role of gd T cells in their breast cancer mouse model in a more detailed way. Why have not they looked at the whole gd T cell compartment? What happens to the overall gd T cell number in WT and KB1P mice after tumor induction? Are there any changes in the CD27- gd T cell compartment?

Response: We have published on the CD27- $\gamma\delta$ T cell compartment recently, see Edwards *et al.*, *J Exp Med* 2023. We found that $V\gamma6+$ cells express constitutive levels of PD-1, while $V\gamma4+$ cells express TIM-3 in response to tumor-derived factors. Blocking antibodies to either PD-1 or TIM-3 fail to modulate cancer progression in the KB1P model. However, these blocking antibodies prevent metastasis in $\gamma\delta$ T cell knockout mice. The current study is focused on CD27+ $\gamma\delta$ T cells.

4. The authors expanded CD27+ gd T cells in vitro and showed that these cells become more cytotoxic and can kill cancer cells in vitro. I suggest to transplant these expanded cells in the KB1P mice to see if they can kill cancer cells in vivo and results in better survival rate of the mice. This will significantly enhance the conclusion regarding the cancer killing activity of these cells and uplift the value of the manuscript. Again, the expansion of CD27+ gd T cells resulting in the increase of their cytotoxic phenotype is independent of Ly6C - arguing if Ly6C is really a good marker to distinguish naïve cells from cytotoxic cells. Therefore, authors should address concern 2 using scRNA-seq to prove that Ly6C can really distinguish cluster 0 from cluster 1.

Response: We performed the experiment as suggested, and it is shown in Figure 4H-K. The data show that Ly6C+ cells are able to control E0771 tumor growth, while Ly6C- cells cannot.

Minor concerns

1. Fig 1B and 1D: no scale bar.

Response: This has been corrected.

2. All FACS plots should have gate frequencies mentioned.

Response: This has been provided in Extended Data Figures 1-5.

Referee #3:

- general summary and opinion

In this manuscript, Wiesheu and colleagues performed single-cell sequencing on murine CD27+ $\gamma\delta$ T cells and found these cells consist of two major populations based on Ly6C expression: a larger population of Ly6C- cells, which exhibit a naive-like phenotype, and a smaller Ly6C+ subset expressing high-levels of cytotoxic genes and show a memory-like phenotype. The authors examined the phenotype of CD27+ and CD27- $\gamma\delta$ T cells in different tissues and among different TCRs ($V\gamma1+$, $V\gamma4+$, $V\gamma1-V\gamma4-$) by flow-cytometry. The anti-tumorigenic activity of $V\gamma1+$ CD27+ $\gamma\delta$ T cells was tested in vivo and in vitro using mammary tumor models. Though the in vivo experiments did not achieve a clear result, the in vitro co-culture of expanded $V\gamma1+$ and tumor cells demonstrated the killing ability of $V\gamma1+$ CD27+ $\gamma\delta$ T cells.

The major conclusion of this study, that CD27+ $\gamma\delta$ T cells can be further discriminated by Ly6C expression as effector/memory-like and naïve/resting-like phenotypes, is convincing. However, similar concepts have been demonstrated by conventional FACS or single-cell sequencing by several previous studies, e.g. (Lombes et al., 2015; Sagar et al., 2020; Sumaria et al., 2017). Thus, it is not very accurate saying 'there is little information on the diversity of CD27+ $\gamma\delta$ T cells in peripheral organs and secondary lymphoid tissue of adult mice at the single-cell level.' Also, the single-cell transcriptome data presented in this study contains less than 500 $\gamma\delta$ T cells only from lung, and the analysis did not go deep (e.g. how these two populations were regulated by TF network). Thus it does not provide much more information on $\gamma\delta$ T biology.

Overall, the manuscript is interesting, but maybe a bit preliminary, and the storyline is not yet fully evolved. A version of this ms is deposited on biorxiv and received robust attention. Each of the three main parts of the ms are interesting, but not very well connected. Part 1 is the scRNA-seq and comparison to human scRNA-seq data, part 2 phenotyping of mouse LN, spleen and lung gd T cells, part 3 the demonstration of anti-tumor activity

Response: As stated in the Introduction, there have been a number of studies reporting on *Ly6c1* or *Ly6c2* expression in CD27+ $\gamma\delta$ T cells by omics methods including works by Sagar *et al.* and Sumaria *et al.* as mentioned above. And while it is true that the Lombes paper demonstrated similar concepts, none of these other studies have addressed the relationship between Ly6C- and Ly6C+ subsets or their functional differences. As such, the novelty of our manuscript lies within its mechanistic description of these subsets. We show for the first time that Ly6C+ cells are terminally differentiated cells with cancer killing abilities supported by IL-27. Our scRNAseq analysis that serves as the foundation of this paper included less than 500 cells, but several studies from other labs support our these observations. We also now provide in-depth transcriptomic data of TCR-defined Ly6C- and Ly6C+ cells (Figure 3) that includes information about transcription factors. More importantly, however, our limited number of cells in scRNAseq analysis provided enough insight into which genes to validate at the protein level (Figure 2); therefore, the cell number is irrelevant. We believe that these new data will satisfy the Reviewer's concerns, as the manuscript is now better connected.

specific major concerns

Introduction:

1. results: For all experiments in this manuscript, please indicate how many individual independent experiments were performed.

Response: This information is provided for all experiments in Figure Legends.

Figure 2

2. Fig 2A: the FACS plot could be gated firstly on CD27+ $\gamma\delta$ T cells, then CCR7 vs Ly6C as a dot plot

Response: This analysis has been provided in updated Fig. 2A.

3. Fig 2C: similarly, better gate on CD27+ $\gamma\delta$ T cells, then Ly6C vs given proteins of interest, not just dot plots vs SSC.

Response: We prefer to leave the representative plots as they stand, as this suggestion is cosmetic and will not change the outcome of the results.

Figure 4

4. Understandably, it is difficult to isolate $\gamma\delta$ T cells from tumors in the mammary glands. However, using cells from pLN, lung, and spleen to demonstrate the roles of $\gamma\delta$ T cells in anti-mammary tumor immunity is problematic.

Response: We show here that CD27+ $\gamma\delta$ T cells are inherently cytotoxic against cancer cells regardless of their origin. We used CD27+ $\gamma\delta$ T cells from lymph nodes and spleen because these organs are enriched for these cells. Obtaining cells from tumors is technically challenging due to their relatively low abundance. Their anti-tumor activity is demonstrated in Fig. 4.

Figure 6

5. The results show that the *in vitro* expansion and gaining of cytotoxicity of CD27+ $\gamma\delta$ T cells is independent to Ly6C expression, which is very interesting and likely to suggest Ly6C- is not a golden standard naïve $\gamma\delta$ T signature.

Response: This is an interesting point, and it is possible that Ly6C- cells include cells at different stages of differentiation. However, our data indicate that Ly6C marks a terminally differentiated subset. We also believe that it is important to separate *in vitro* and *in vivo* results when interpreting these data. *In vivo*, Ly6C- and Ly6C+ cells isolated directly from mice are phenotypically distinct, whereas culturing cells with IL-2 and IL-15 *in vitro* makes them more phenotypically similar – at least for the proteins we analyzed, bearing in mind that we haven't done full transcriptome or proteome analysis of *ex vivo* expanded cells. Both cytokine-expanded populations can kill cancer cells. However, expanded Ly6C+ cells still maintain a superior killing ability over expanded Ly6C- cells (Fig. 4G). When transferred into tumor-bearing mice, only Ly6C+ cells can control tumor growth (Fig. 4H-K). Therefore, Ly6C+ cells mark a more evolved effector subset.

minor concerns

Figure 1

6. Many genes authors picked up to explain the differences between cluster 0 & 1, e.g. Cd160, Ncr1, Gzma, S1pr1, and ribosomal genes, were not shown in Fig 1B-D, nor their detailed expression data (logFC, p-value, etc) were provided in Tables.

Response: These data are now provided in Table 1. Fig. 1 focuses on the top 10 differentially expressed genes only.

7. Fig 1E,F: the analysis to calculate the modular expression of naïve and cytotoxic signatures on human dataset is interesting, the result its self is convincing. The cytotoxic & naïve T cell signature genes are largely conserved is not very surprising, however, it is too preliminary to suggest this "strong rationale to investigate the function of anti-tumour $\gamma\delta$ T cells from mice to inform human $\gamma\delta$ T cell biology".

Response: We are happy that the Reviewer agrees with our conclusion about the similarity between species. Given that the phenotype of mouse $\gamma\delta$ T cells is highly analogous to human $\gamma\delta$ T cells, we believe that mouse experiments can model some aspects of human $\gamma\delta$ T cells. However, we have removed this statement because we do not have the word space to extrapolate on the strengths and weaknesses of using mice to represent humans.

Figure 2

8. Fig 2C&D: as pointed out above, 3 out of the 4 markers used here were not shown in Fig 1B-D, making readers unable to compare the transcription level & expression level.

Response: This has been provided in Table 1.

Figure 5

9. V γ 1 depletion appears ok, however it would be useful to add proper controls and references for V γ 1 depletion

Response: These data have been removed as discussed above in response to Reviewer #1.

10. Fig 5C&D depict the Ly6C expression divided by TCR, which is not directly related to anti-tumor immunity, this part is better to be in earlier figures.

11.

Response: We agree and these data are now presented in Fig. 3.

References

- Lombes, A., Durand, A., Charvet, C., Rivière, M., Bonilla, N., Auffray, C., Lucas, B., and Martin, B. (2015). Adaptive Immune-like γ/δ T Lymphocytes Share Many Common Features with Their α/β T Cell Counterparts. *J. Immunol.* 195, 1449-1458.
- Sagar, Pokrovskii, M., Herman, J.S., Naik, S., Sock, E., Zeis, P., Lausch, U., Wegner, M., Tanriver, Y., Littman, D.R., et al. (2020). Deciphering the regulatory landscape of fetal and adult γ/δ T-cell development at single-cell resolution. *EMBO J.*
- Sumaria, N., Grandjean, C.L., Silva-Santos, B., and Pennington, D.J. (2017). Strong TCR γ/δ Signaling Prohibits Thymic Development of IL-17A-Secreting γ/δ T Cells. *Cell Rep.* 19, 2469-2476.

Dear Seth,

Congratulations on a great revision! Overall, the referees have been positive. However, referee 1 has raised a few concerns that we ask you to non-experimentally address in a revised version of your manuscript. When you submit your revised version, please also take care of the following editorial items and add this also to your point-by-point response:

1. Please include an author checklist found on our author website
2. Please rename extended data figures as Figure EV1, etc. updating this in the legends and main manuscript.
3. Please provide up to five keywords, which may or may not appear in the title, should be given in alphabetical order, below the abstract, each separated by a slash (/).
4. Please add all authors to eJP online submission system.
5. The data availability section appears twice in the manuscript.
6. Please update the order of sections to: Abstract / Introduction / Results / Discussion / Materials and Methods / Acknowledgements / Disclosure and competing interests statement / References / Figure legends / Tables and their legends / Expanded View Figure legends
7. Please merge the funding statement with acknowledgements and ensure that all funding should be entered into eJP online.
8. Please remove the author contribution section from the main manuscript.
9. On p41, what is listed as author contributions should be renamed to "Disclosure and competing interests statement".
10. Please rename Tables 2 and 3 to Dataset EV1 and EV2 and both need legends added to the files in a separate sheet.
11. We require the publication of source data, particularly for electrophoretic gels and blots and graphs, with the aim of making primary data more accessible and transparent to the reader. It would be great if you could provide me with a PDF file per figure that contains the original, uncropped and unprocessed scans of all or key gels used in the figure or for graphs, an Excel spreadsheet with the original data used to generate the graphs. The PDF files should be labeled with the appropriate figure/panel number, and should have molecular weight marker; further annotation could be useful but is not essential. The PDF files will be published online with the article as supplementary "Source Data" files.
12. We include a synopsis of the paper (see <http://emboj.embopress.org/>). Please provide me with a general summary statement and 3-5 bullet points that capture the key findings of the paper.
13. We also need a summary figure for the synopsis. The size should be 550 wide by 200-440 high (pixels). You can also use something from the figures if that is easier.
14. Please remove the one-sentence summary.
15. The table on pages 24-26 needs a title (e.g. Table 1), callout and legend added, and it should be moved after the main figure legends.
16. Panel A should be removed for Fig 8 as there is only one panel.
17. Please provide the specific URL for E-MTAB10677 and E-MTAB-13897 datasets in the data availability section. Please also provide the reviewer access code for E-MTAB-13897.
18. Please note that a separate 'Data Information' section is required in the legends of figures 2b-f; 4a-g; 5a-d; 6c-f, h-i.
19. Please indicate the statistical test used for data analysis in the legends of figures 3f-g
20. Please note that in figures 4g; 7a, f-g; there is a mismatch between the annotated p values in the figure legend and the annotated p values in the figure file that should be corrected.
21. Please provide N in the legend of figure 1c.
22. Please describe the nature of N in the legends of fig3a-b.

23. Please define the error bars in the legends of fig3a-b, 7a, 7d-h, 7j, EV5b.

Thank you for the opportunity to consider your work for publication, I look forward to your revision.

Warm wishes,
Kelly

Kelly M Anderson, PhD
Editor, The EMBO Journal
k.anderson@embojournal.org

- a complete author checklist, which you can download from our author guidelines
(<https://www.embopress.org/page/journal/14602075/authorguide>).

- Expanded View files (replacing Supplementary Information)

Further information is available in our Guide For Authors: <https://www.embopress.org/page/journal/14602075/>

authorguide

Referee #1:

The points I've raised have been sufficiently addressed by the authors.

I suggest two minor changes;

1) Fig 4 J/K: please tone down the statement that "cytotoxic phenotype of CD27+Ly6C+ gd T cells equates to superior cancer-killing ability" (line 245) in vivo. There is no significant difference between CD27+Ly6C- and CD27+Ly6C+ cells and no analysis of numbers, activation exhaustion, senescence markers is shown to confirm that only cytotoxicity is different between the two subsets. Furthermore, only 2 out of 3 mice have prolonged survival and no beneficial effects of adoptive transfer were seen in the other mice making the choice of model system questionable.

2) Fig 6I tetramer staining: a small change in tetramer length can only be interpreted with expression levels of telomerase. Has this been assessed in the respective subsets?

Referee #2:

The manuscript has been significantly improved during the review process and the authors have addressed all my concerns. This is a nice and significant piece of work. Congratulations!

Referee #3:

The authors have made an effort to reply to all concerns of the three reviewers and have successfully improved the ms.

Referee #1:

The points I've raised have been sufficiently addressed by the authors.

I suggest two minor changes;

1) Fig 4 J/K: please tone down the statement that "cytotoxic phenotype of CD27+Ly6C+ gd T cells equates to superior cancer-killing ability" (line 245) *in vivo*. There is no significant difference between CD27+Ly6C- and CD27+Ly6C+ cells and no analysis of numbers, activation exhaustion, senescence markers is shown to confirm that only cytotoxicity is different between the two subsets. Furthermore, only 2 out of 3 mice have prolonged survival and no beneficial effects of adoptive transfer were seen in the other mice making the choice of model system questionable.

Response: We are delighted to hear that all the Referee's points are addressed. We have modified the disputed sentence to better reflect the results (lines 240-241). Please note that 5-6 mice were used per group in the E0771 model. Survival was not changed by adoptive transfer of naïve CD8 T cells or CD27+Ly6C⁻ $\gamma\delta$ T cells into tumor-bearing mice, whereas median survival was extended when tumor-bearing mice were injected with CD27+Ly6C⁺ $\gamma\delta$ T cells as compared to PBS control. Perhaps, more than 4 injections or increased injected cell numbers would have extended survival even further; however, these experiments are technically and logistically challenging so we did not attempt other alterations to the protocol.

2) Fig 6I tetramer staining: a small change in tetramer length can only be interpreted with expression levels of telomerase. Has this been assessed in the respective subsets?

Response: We believe that the Referee is referring to telomeres rather than tetramers, because no tetramer staining was performed in this study. We appreciate the nice suggestion – we did not measure telomerase expression; although, we agree that this should be done for future studies.

Referee #2:

The manuscript has been significantly improved during the review process and the authors have addressed all my concerns. This is a nice and significant piece of work. Congratulations!

Response: We greatly appreciate the Referee's use of the phrase, "...significant piece of work." Thank you!

Referee #3:

The authors have made an effort to reply to all concerns of the three reviewers and have successfully improved the ms.

Response: Thank you!

Dear Seth,

Congratulations on an excellent manuscript, I am pleased to inform you that your manuscript has been accepted for publication in the EMBO Journal. Thank you for your comprehensive response to the referee concerns and for providing detailed source data. It has been a pleasure to work with you to get this to the acceptance stage.

I will begin the final checks on your manuscript before submitting to the publisher next week. Once at the publisher, it will take about 3 weeks for your manuscript to be published online. As a reminder, the entire review process, including referee comments and your point-by-point response, will be available to readers.

I will be in touch throughout the final editorial process until publication. In the meantime, I hope you find time to celebrate!

Warm wishes,
Kelly

Kelly M Anderson, PhD
Editor, The EMBO Journal
k.anderson@embojournal.org

Please note that you will be contacted by Springer Nature Author Services to complete licensing and payment information.
